# DISTRIBUTION-FREE DATA UNCERTAINTY FOR NEURAL NETWORK REGRESSION

**Domokos M. Kelen**[*†]     **Ádám Jung**[†]     **Péter Kersch**[§]     **András A. Benczúr**[†‡]

## ABSTRACT

Quantifying uncertainty is an essential part of predictive modeling, especially in the context of high-stakes decision-making. While classification output includes data uncertainty by design in the form of class probabilities, the regression task generally aims only to predict the expected value of the target variable. Probabilistic extensions often assume parametric distributions around the expected value, optimizing the likelihood over the resulting explicit densities. However, using parametric distributions can limit practical applicability, making it difficult for models to capture skewed, multi-modal, or otherwise complex distributions. In this paper, we propose optimizing a novel nondeterministic neural network regression architecture for loss functions derived from a sample-based approximation of the continuous ranked probability score (CRPS), enabling a truly distribution-free approach by learning to sample from the target's aleatoric distribution, rather than predicting explicit densities. Our approach allows the model to learn well-calibrated, arbitrary uni- and multivariate output distributions. We evaluate the method on a variety of synthetic and real-world tasks, including uni- and multivariate problems, function inverse approximation, and standard regression uncertainty benchmarks. Finally, we make all experiment code publicly available[1].

## 1 INTRODUCTION

With the continued development of machine learning, more and more industries consider using advanced predictive modeling to influence automated or human made decisions. Modern methodologies such as deep learning offer unparalleled accuracy for modeling complex real-world processes, with the potential to greatly enhance efficiency in many applications. Uncertainty quantification (UQ) is an essential part of predictive modeling, especially in applications where the prediction has the potential to influence high-stakes decision making. A large body of research deals with predicting the target value along with its uncertainty, such as Bayesian Neural Networks (BNN).

In a prediction task, uncertainty comes from multiple different sources. BNN-s are primarily concerned with *epistemic* (or knowledge) uncertainty, the uncertainty of how a model should predict the future based on the limited data we have at our disposal. In use-cases where data is limited, such approaches are invaluable for acknowledging potential blind spots in the modeling process.

However, in many industry applications, data is abundant. Still, a different kind of uncertainty remains even when data is plentiful: *aleatoric* (or data) uncertainty arises from constraints in the problem setup, where the model lacks information to make an exact prediction. Most tasks involve key pieces of information missing, factors which influence the target, yet ones we cannot provide to the model as inputs. Either due to inherent randomness or unobservable latent variables, in most real-world scenarios it is theoretically impossible to always predict the exact target value. As an inherent constraint, aleatoric uncertainty cannot be reduced with additional data of the same kind.

Classification tasks are capable of expressing aleatoric uncertainty by design in the form of class probabilities. However, in the case of regression, the models traditionally only output a single value, or a simple parametric distribution at best. An *expectation* of 10mm of rainfall could indicate relative certainty of a rainy day with actual 10mm of rainfall, but could also indicate 10% chance of a deadly

---

[*]Emails: {kdomokos, jungadam, benczur}@info.ilab.sztaki.hu, peter.kersch@ericsson.com
[†]HUN-REN SZTAKI     [§]Ericsson Hungary     [‡]Széchenyi University, Győr, Hungary

[1]https://github.com/proto-n/torch-naut

storm with 100mm of rain. In such cases, even a heteroscedastic distributional regression model with a simple unimodal output lacks the nuance required for informed decision making.

In this work, we propose a novel nonparametric way to model aleatoric uncertainty, relaxing the usual constraints of simple parametric distributions such as unimodality, and enabling the approximation of arbitrary distributions. We introduce new nondeterministic neural network architectures to generate samples from the aleatoric distribution of the target, and train the resulting models by drawing upon the theory of scoring rules, originally developed for evaluating probabilistic forecasts in fields such as meteorology and economics. We derive a novel, sample-based loss function based on an unbiased $\mathcal{O}(n \log n)$ formulation of the empirical *continuous ranked probability score* (CRPS), a strictly proper scoring rule for regression, describing both unweighted and weighted variants.

The resulting method is capable of learning arbitrary probability distributions from sample data in a well-calibrated manner, no longer being restricted to simple parametric uni-modal distributions, at the same time remaining conceptually simple and easy to implement. Further, it generalizes for multivariate regression, and is capable of learning the joint distribution of hundreds of output values. We evaluate our approach on multiple tasks, including synthetic uni- and multivariate examples, the inverse problem of MNIST classification, and also on the standard UCI regression uncertainty benchmark. The method consistently exhibits favorable behavior in practice, making it an appealing choice for modeling aleatoric uncertainty. Our contributions can be summarized as follows:

- We propose multiple novel architectures of nondeterministic neural networks for producing samples from the aleatoric target distribution with an efficient optimization procedure.
- We derive a loss function for optimizing neural networks using a novel unbiased empirical approximation of the CRPS, describing the loss for both unweighted and weighted samples.
- Finally, we extensively test the resulting method, comparing it against prior work such as BNN-s (Gawlikowski et al., 2023) or Mixture Density Networks (MDN) (Bishop, 1994).

**Scaling.** Our approach scales well to larger network sizes. While part of the model is evaluated multiple times, the incurred cost scales with target rather than input complexity, see Appendix C.8. Our experiments show that our models can represent a wide range of complex distributions, using a relatively small nondeterministic part considering modern deep learning hardware and model sizes.

## 2 BACKGROUND

### 2.1 TYPES OF UNCERTAINTY

In prediction tasks, different sources of uncertainty can be distinguished. This section briefly summarizes the two main types we focus on; for a general overview, see (Gawlikowski et al., 2023).

Generally, the data we record and collect comes from a real-world process that is only partially observed. In regression, the target value (i.e., the value that we aim to predict) can be modeled as a random variable $Y : \Omega \to \mathbb{R}$, where elementary events $\omega \in \Omega$ are assumed to represent the true state of the world. Although we are unable to directly observe $\omega$, we can observe a number of feature variables $X : \Omega \to \mathbb{R}^m$, which give us partial information about $\omega$. The classic regression task is then defined as approximating the conditional expectation $\mathrm{E}[Y \mid X = x, D]$, where the data $D$ is a sample of past values of $(X, Y)$. In multivariate regression, $Y$ takes on values from $\mathbb{R}^k$. Uncertainty quantification extends the regression task so that we are no longer simply interested in the conditional expectation, but rather the entire conditional distribution of $y$, i.e., $p(y \mid X = x, D)$.

Aleatoric uncertainty (Kendall & Gal, 2017) arises from $X$ giving us limited information about $\omega$. We can express aleatoric uncertainty through the conditional cumulative distribution function as

$$F_{Y|X}(y_0) = \int_\Omega \mathbb{1}\{Y(\omega) < y_0\} \, p(\omega \mid X) \, \mathrm{d}\omega. \tag{1}$$

Many constraints can be modeled through aleatoric uncertainty, for example measurement error, or uncertainty due to unobservable latent variables that affect $Y$. This kind of uncertainty *cannot* be reduced with more data (i.e., a larger sample $D$), as the uncertainty arises from constraints in the problem setup. In other words, aleatoric uncertainty is the uncertainty that remains even if we assume to have perfect knowledge of the relationships between $X, Y$, and $\omega$.

Epistemic uncertainty on the other hand stems the fact that we have no perfect knowledge of the relationship of $X, Y$, and $\omega$. Equation (1) is described in idealistic terms, and in fact neither term of the integration is known in practice, as our only information about $X$ and $Y$ comes from the sample $D$. Epistemic uncertainty is then modeled as the uncertainty of selecting $f$ such that $f(y) \approx p(y \mid X = x, D)$, with larger dataset $D$ implying less uncertainty. The majority of uncertainty research is focused on epistemic uncertainty, for example the research of BNN-s, see Section 2.2.

In classification, aleatoric uncertainty is expressed through class probabilities, while regression traditionally focuses on point estimates like the expectation. In probabilistic regression, aleatoric regression uncertainty is often modeled using parametric distributions, with the normal distribution being a particularly common choice (Jospin et al., 2022; Abdar et al., 2021; Magris & Iosifidis, 2023). However, parametric distributions are limited in their expressive power, and the wrong choice of distribution may undermine the entire learning process. Nevertheless, little attention is usually paid to the representation of the distribution of $Y$, with most uncertainty-related work either simply assuming the existence of a likelihood function or explicitly predicting the parameters of a normal distribution. This oversight persists despite the fact that that arbitrary-shaped aleatoric distributions can arguably be learned even when only a single value $y$ is recorded for each input $X$, by leveraging the same implicit assumptions of smoothness that supervised learning generally relies on.

## 2.2 BAYESIAN NEURAL NETWORKS

Bayesian neural networks (or BNN-s) (Blundell et al., 2015) are nondeterministic neural networks whose stated goal is to quantify epistemic uncertainty. Using the terminology of Section 2.1, BNN-s model the conditional density of $Y$ through a neural network parameterized by $\theta \in \Theta$, i.e., $f_\theta(y) \approx p(y \mid X = x, D)$. Epistemic uncertainty is then expressed as the uncertainty of $\Theta$, i.e.,

$$p(y \mid X = x, D) = \int_\Theta p(y \mid \theta, X = x) \, p(\theta \mid D) \, d\theta, \tag{2}$$

where the first term being integrated is the aleatoric likelihood, in practice output by the neural network after having sampled $\theta$ according to the second term. A large body of research deals with the theory of the optimization of BNN-s, see (Gawlikowski et al., 2023) for an overview.

Each evaluation of a BNN yields a separate aleatoric uncertainty distribution, while epistemic uncertainty is captured by the nondeterminism of the network, with subsequent runs yielding different aleatoric uncertainty distributions sampled from the epistemic uncertainty distribution. In other words, epistemic uncertainty can be interpreted as the uncertainty of aleatoric uncertainty. Accordingly, they are sometimes referred to as first- and second-order uncertainties (Bengs et al., 2023).

Generally, aleatoric uncertainty is expressed in the form of a parametric distribution, while epistemic uncertainty is usually treated as nonparametric: while the distribution of the weights $\theta$ is individually assumed to be parametric, the resulting distribution of the output of the whole network can take arbitrary shapes. In this paper, we propose modeling both kinds as nonparametric distributions.

## 2.3 PROPER SCORING RULES

When a model outputs a distribution, scoring rules can be used to quantify the accuracy of predictions based on a sample from the target, i.e., they are analogous to metrics like RMSE and AUC but specifically for probabilistic predictions. Generally, a scoring rule S takes a predicted distribution $P$ and a sample $y$ from the target, and assigns a score $S(P, y)$. Denoting $E_{Y \sim Q}[S(P, Y)]$ as $S(P, Q)$, a scoring rule is *proper* if $S(Q, Q) \leq S(P, Q)$ for all $P$, and *strictly proper* if equality implies $P = Q$ and vice versa. Strictly proper scoring rules encourage predictors to provide honest predictive distributions that accurately reflect the target's distribution (Gneiting & Raftery, 2007). An example is the logarithmic score (Good, 1952), defined as $LogS(F, y) = -\log(f(y))$, where $f$ is the density of $F$, more commonly known in machine learning literature as the negative log-likelihood.

A strictly proper scoring rule for regression is the *continuous ranked probability score* (CRPS):

$$\mathrm{CRPS}(F, y) = \int_{\mathbb{R}} (F(z) - \mathbb{1}\{y \leq z\})^2 \, dz, \tag{3}$$

where $F$ is the cumulative distribution function (CDF) of the predicted distribution and $y$ is the target sample. The CRPS can be analytically computed for a number of parametric distributions, see Jordan et al. (2019) for a comprehensive list.

## 2.4 EMPIRICAL CRPS

To approximate Equation (3) in practice, we can compute the formula substituting $F$ for the empirical CDF of a sample. As described by (Gneiting & Raftery, 2007), an equivalent formulation (Baringhaus & Franz, 2004; Székely & Rizzo, 2005) of Equation (3), assuming $\mathrm{E}[F] < \infty$, is

$$\mathrm{CRPS}(F, y) = \mathrm{E}\,|Y - y| - \frac{1}{2}\,\mathrm{E}\,|Y - Y'|, \text{ where } Y, Y' \sim F,\ Y \perp\!\!\!\perp Y'. \tag{4}$$

When evaluating Equation (4) for the empirical CDF ($F_{\mathrm{ECDF}}$) (Krüger et al., 2021) of a $(\hat{y}_1, \ldots, \hat{y}_n)$ sample distributed according to $F$, the resulting empirical score $\mathrm{CRPS}(F_{\mathrm{ECDF}}, y)$ is given by

$$\frac{1}{n}\sum_{i=1}^{n}|y - \hat{y}_i| - \frac{1}{2n^2}\sum_{i=1}^{n}\sum_{j=1}^{n}|\hat{y}_i - \hat{y}_j| = \frac{2}{n^2}\sum_{i=1}^{n}\left(\hat{y}_{(i)} - y\right)\left(n\mathbb{1}\{y < \hat{y}_{(i)}\} - i + \frac{1}{2}\right). \tag{5}$$

While the naive formula on the left hand side is $\mathcal{O}(n^2)$ complexity, the right hand side gives a relatively well-known (Murphy, 1970; Jordan, 2016; Laio & Tamea, 2007; Jordan et al., 2019) algebraically equivalent $\mathcal{O}(n\log n)$ formula, with $(\hat{y}_{(1)}, \ldots, \hat{y}_{(n)})$ denoting the ordered sample statistic.

## 2.5 MIXTURE DENSITY NETWORKS

While most uncertainty-related work does not investigate distribution-free approaches for aleatoric regression uncertainty, one exception is Mixture Density Networks (MDN) (Bishop, 1994). MDN expresses the density as a mixture of a fixed number Gaussians. As the number of components tends to infinity, a Gaussian mixture is capable of approximating any distribution, therefore we consider MDN one of the strongest prior art in this field, also belonging to the class of models capable of learning aleatoric uncertainty in a distribution-free manner. However, MDN models often have practical limitations. In our experiments, using our own robust implementation we achieve much stronger scores for MDN then previously reported, see Appendix C.2 and the source code for details.

## 3 RELATED WORK

The literature of uncertainty quantification is vast, with the majority of works focusing on quantifying epistemic uncertainty, for example the BNN (Blundell et al., 2015), or its many enhancements (Sun et al., 2017; Gawlikowski et al., 2023). The UCI Benchmark (Hernández-Lobato & Adams, 2015) has become the standard benchmark to measure regression uncertainty, with many papers reporting RMSE and (negative) log-likelihood (NLL) results along with standard error. See Appendix A.1 for a list. While certain nonparametric regression approaches, e.g., some variants of as Gaussian Process Regression (Rasmussen, 2003; Sendera et al., 2021), are capable of modeling the output in a distribution-free manner, the area of distribution-free approaches for neural network-based aleatoric regression uncertainty is relatively under-researched. Such methods, for example MDN (Bishop, 1994), seem to be under-utilized and under-reported in uncertainty-related literature, with most surveys barely mentioning or omitting the topic, see Appendix A.2 for a small survey. MDN itself rarely appears as baseline. We are only aware of (El-Laham et al., 2023) on the UCI benchmark, however with substantially lower scores. See Appendix C.2 for more information.

Minimizing proper scoring rules to learn is common practice, either implicitly or explicitly. Both maximum likelihood for regression and cross entropy for binary classification coincide with minimizing the logarithmic score (Good, 1952), also a main component of variational inference for BNN-s (Blundell et al., 2015). Maximum likelihood estimation with an assumed Gaussian likelihood coincides with the Dawid-Sebastiani score (Dawid & Sebastiani, 1999; Gneiting & Raftery, 2007; Czado et al., 2009). Examples of explicitly using scoring rules to learn include Lakshminarayanan et al. (2017); Bengs et al. (2023); Vahidi et al. (2023); Bouchacourt et al. (2016). Prior work using the CRPS for learning often does so to learn combination weights (Thorey et al., 2017; Berrisch & Ziel, 2023; van der Meer et al., 2024). An analytic expression of the CRPS for the Gaussian is used for optimization in Rasp & Lerch (2018); Gebetsberger et al. (2018).

The biased sample-based $\mathcal{O}(n\log n)$ CRPS formula is well-known (Murphy, 1970; Jordan, 2016; Laio & Tamea, 2007; Jordan et al., 2019), and the bias of the $\mathcal{O}(n^2)$ formula has also been noted before (Ferro et al., 2008; Thorey et al., 2017), however to our knowledge we are the first to describe

the unbiased $\mathcal{O}(n \log n)$ formula. Weighted CRPS versions exist (Gneiting & Ranjan, 2011; Allen, 2023; Taggart, 2023; Pic et al., 2023), however in the sense of weights over the range of $Y$ to emphasize certain target values. In contrast, we introduce a *weighted-sample*-based approximation for the CRPS and, to our knowledge, are first to describe both the $\mathcal{O}(n^2)$ and the $\mathcal{O}(n \log n)$ formulas.

Non-gaussian output distributions beyond MDN have been studied before, e.g., the method of (Depeweg et al., 2018) is capable of representing complex distributions, including multi-modal ones. In the context of generative modeling, related concepts are, e.g., Normalizing Flows (Ardizzone et al., 2018), and *Maximum Mean Discrepancy* (MMD) (Li et al., 2015; Dziugaite et al., 2015; Du et al., 2023; Hertrich et al., 2024; Cui et al., 2020), which has a close connection to *kernel scoring rules* (such as the CRPS) (Zawadzki & Lahaie, 2015; Schölkopf, 2000; Sejdinovic et al., 2012).

Closest to our approach is Bouchacourt et al. (2016) (DISCO Nets), a probabilistic method for hand-pose estimation. They also use injected random noise for nondeterminism (as in Section 4.1), and derive a dissimilarity coefficient (Rao, 1982) based loss function that reduces to the Energy Score as special case (see Section 4.5), which the authors also highlight. Developed independently, our work focuses on optimizing the CRPS, whose multivariate generalization also coincides with the Energy Score. Our work extends beyond DISCO Nets through an $\mathcal{O}(n \log n)$ loss formula instead of $\mathcal{O}(n^2)$, weighted and multi-head architectures, explicit handling of epistemic uncertainty, and further adaptations for traditional regression. We believe the connections between the two approaches serve to further emphasize the broader applicability of the shared ideas behind both.

# 4 LEARNING TO SAMPLE FROM THE ALEATORIC DISTRIBUTION

We propose optimizing neural networks using $\mathrm{CRPS}(F_{\mathrm{ECDF}}, y)$ from Equation (5) as a loss function. By taking multiple samples from a nondeterministic neural network, we leverage automatic differentiation to backpropagate the loss across multiple evaluations at the same time. Note that since the ordered sample is a sufficient statistic, the *sort* operation does not need to be backpropagated.

The effects of Equation (5) interpreted as a loss function are quite intuitive. The first term is equivalent to the mean absolute error (MAE), forcing the network to center its predictions around the median of the target distribution. The second term is being maximized due to its negative sign, forcing the distribution to spread out, increasing its variance. With the $\frac{1}{2}$ coefficient, balance is found exactly when the predicted distribution follows the target distribution (Gneiting & Raftery, 2007).

## 4.1 FAST UNBIASED SAMPLE-BASED CRPS LOSS

The formula for the $\mathcal{O}(n \log n)$ sample-based CRPS of Equation (5) is relatively well-known. Less widely known is the fact that the formulas of Equation (5) are biased, not unlike the naive estimate for empirical variance. As described by Ferro et al. (2008), Equation (5) underestimates the second term because the diagonal is always constant zero, unlike in the original of Equation (4). As a result, the naive formula underestimates prediction variance, making the model underconfident. An unbiased version (Ferro et al., 2008; Thorey et al., 2017) can be calculated by using the coefficient $\frac{1}{2n(n-1)}$ instead of $\frac{1}{2n^2}$. We derive the corresponding modification of the $\mathcal{O}(n \log n)$ formula as

$$\mathrm{CRPS}'(F_{\mathrm{ECDF}}, y) = \frac{2}{n(n-1)} \sum_{i=1}^{n} \left(\tilde{y}_{(i)} - y\right) \left((n-1)\mathbb{1}\{y < \tilde{y}_{(i)}\} - i + 1\right), \quad (6)$$

where $\mathrm{CRPS}'$ denotes the unbiased version of the empirical CRPS, see Appendix B.4 for proof.

## 4.2 OPTIMIZING NEURAL NETWORKS FOR THE EMPIRICAL CRPS

While any kind of nondeterministic neural network with sufficient expressive power should work in theory, we see no point in, e.g., sampling different weights for the network with each evaluation, as BNN-s do. Instead, we propose injecting the randomness through independent standard normal random samples $\varepsilon$ concatenated to the input or a hidden layer activation, see Figure 1a. Exploring other nondeterminism injection methods is an interesting research topic, however is out of scope.

Using the terminology of Section 2.1, the resulting network can be interpreted as follows. First, the network transforms $X$ into a latent representation $\omega_x$ revealing partial information about $\omega$.

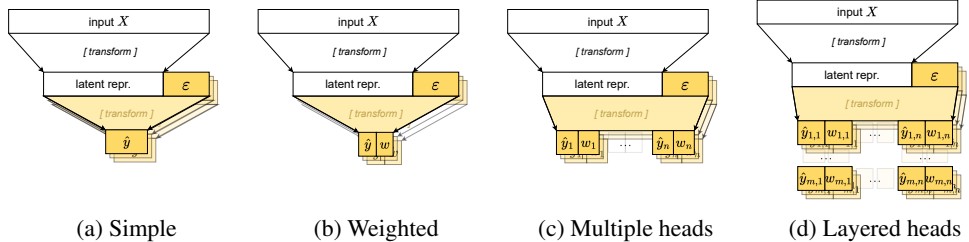

Figure 1: Illustration of nondeterministic neural network architectures. The top (empty) part of each network is evaluated once, responsible for transforming the input. The yellow (solid) parts of each network are evaluated multiple times, with $\varepsilon$ being randomly sampled each time.

Second, we sample $\varepsilon$ in place of the unobservable latent variables that, together with X, are assumed to identify $\omega$ with certainty. Finally, the second part learns the deterministic mapping $(\omega_x, \varepsilon) \mapsto y$ which draws samples from the target distribution, with random $\varepsilon$ values sampled in each run. In practice, only the second part needs to be evaluated multiple times. For further efficiency, the network can be implemented to compute multiple samples in a single batch, with output of shape $[b \times n \times k]$, where $b$ is batch size, $n$ is the number of samples, and $k$ is the dimension of $Y$.

In theory, such a network can learn to produce samples from arbitrary probability distributions, and CRPS being a strictly proper scoring rule guarantees that the global optimum corresponds to the real aleatoric distribution under ideal conditions (i.e., assuming infinite data and MLP-s to be universal function approximators). Ultimately, the described setup provides a truly nonparametric way to learn aleatoric uncertainty for neural network regression.

## 4.3 Weighted CRPS Loss

The simple architecture described in Section 4.2 needs to model distributions by not only covering the full range of possible outputs depending on the injected randomness, but also making sure that potential outputs get sampled proportionally from the target distribution. Consequently, output regions with small probability get sampled infrequently during training, possibly leading to the model being unable to learn them in detail. Further, neural networks have an inherent bias towards learning smooth mappings from input to output, leading to the learned distribution possibly becoming overly smoothed when the network is limited in its capacity or cannot be trained until convergence (i.e., early stopping is used), both of which are common in practical applications.

An implicit bias towards smooth functions can be useful in certain applications, however it is also a limitation of the simple unweighted architecture. To improve the expressivity of the network, we introduce the weighted-sample-based CRPS loss (weighted CRPS for short), along with a modified architecture which produces a weighted sample as its output. By allowing a weighted sample, we are essentially allowing the model to draw samples $(\hat{y}_i)$ from distribution $Q'$ instead of $Q$, along with the change-of-measure coefficients $w = \frac{\mathrm{d}Q}{\mathrm{d}Q'}$ (also known as likelihood-ratio or importance weight).

We derive the unbiased approximation of the CRPS in Appendix B.1 from a weighted sample as

$$\mathrm{CRPS}_w(F^w_{\mathrm{ECDF}}, y) = \frac{1}{n}\sum_{i=1}^{n} w_i|y-\hat{y}_i| - \frac{1}{2n(n-1)}\sum_{i=1}^{n}\sum_{j=1}^{n} w_i w_j|\hat{y}_i-\hat{y}_j|, \quad (7)$$

where $w_i = \frac{\mathrm{d}Q}{\mathrm{d}Q'}(\hat{y}_i)$, and $F^w_{\mathrm{ECDF}}$ denotes the empirical CDF of the weighted sample $(\hat{y}_i, w_i)$. We also derive an algebraically equivalent unbiased $\mathcal{O}(n \log n)$ formula, analogous to Equation (6), as

$$\mathrm{CRPS}'_w(F^w_{\mathrm{ECDF}}, y) = \frac{2}{n(n-1)}\sum_{i=1}^{n} w_{(i)}\big(\hat{y}_{(i)}-y\big)\left((n-1)\,\mathbb{1}\big\{y<\hat{y}_{(i)}\big\}-s_i+\frac{W-n+w_{(i)}+1}{2}\right), \quad (8)$$

denoting $W = \sum_{i=1}^{n} w_{(i)}$ and $s_i = \sum_{j=1}^{i} w_{(j)}$. See Appendix B.2 and B.3 for proof.

## 4.4 Weighted sampling architectures

While the weighted loss may in itself enhance model expressivity, its more significant impact is enabling advanced network architectures. A simple architecture must evaluate the non-deterministic

network component to generate new samples, requiring non-negligible compute time for larger sample sizes during both training and prediction. The weighted formula improves sampling efficiency by allowing the network to produce multiple weighted samples in each evaluation (see Figure 1c). Without weighting, such a network would be problematic, as all samples from the same evaluation would share the same frequency in the overall sample.

The weighted formula can leverage multiple such *sampling heads* while still ultimately being capable of representing arbitrary distributions. This increases expressivity, as each head samples from distinct parts of the output distribution, modeling simpler distributions individually. However, early stopping can cause transitions between heads to be non-seamless. A convenient solution is to apply the loss function separately to multiple sets of heads (see Figure 1d), resulting in heads with overlapping ranges. This approach resembles an in-network ensemble, however to avoid confusion with traditional ensembling, we refer to it as *layered sampling heads*. The synthetic task in Section 5.1 serves as a good visual illustration of the effects of different sampling architectures.

### 4.5 MULTIVARIATE REGRESSION

The formulas described so far only apply to univariate targets. Fortunately, a multivariate generalization of the CRPS exists, called the *energy score* (Gneiting & Raftery, 2007; Jordan et al., 2019):

$$\text{ES}(F, y) = \frac{1}{n} \sum_i \|y - \tilde{y}_i\|_2 - \frac{1}{2n^2} \sum_i \sum_j \|\tilde{y}_i - \tilde{y}_j\|_2, \tag{9}$$

with the same caveats as Equation (5) regarding complexity and bias. While unbiased and weighted versions can be calculated analogously, the $\mathcal{O}(n \log n)$ formula is only valid for the univariate case. Potential ways to speed up the calculation do exist, see for example Hertrich et al. (2024). However, investigating the multivariate case in detail is out of scope for this work. Nevertheless, we do conduct experiments on multivariate problems using Equation (9) in its quadratic form, see Section 5.2.

### 4.6 EPISTEMIC UNCERTAINTY

The primary goal of our approach is to learn the aleatoric uncertainty of the target variable $Y$ in a frequentist manner, without the use of Bayesian techniques. Although our method could be extended by mixing samples from a prior distribution into the training data as a way to mimic Bayesian priors, as it is sometimes done for example with out-of-distribution data (Malinin, 2019; Lakshminarayanan et al., 2017; Chen et al., 2018; Hafner et al., 2020), we do not attempt such approaches in this paper. Still, the benchmark used in Section 5.3 involves a large number of relatively small datasets, making it essential to account for epistemic uncertainty. Fortunately, the ensemble-based method of Lakshminarayanan et al. (2017) combines well with our proposed approach, as the output of multiple independently trained models can be mixed by simply pooling the samples.

### 4.7 LIKELIHOOD ESTIMATION

Since our method represents the output distribution through producing samples, it is very easy to estimate, e.g., percentiles. On the other hand, without an explicit density, it is nontrivial to evaluate likelihood at a given point. However, we still need to do so to compare against prior work, which generally evaluate methods using the negative log-likelihood metric (see Section 5).

In general, a sample-based unbiased estimator for the likelihood can not exist (Rosenblatt, 1956). However, the literature on *nonparametric density estimation* (Izenman, 1991) is vast, with many methods not only guaranteeing convergence, but providing useful estimates at smaller sample sizes. Given the ease of sampling in our case, the lack of explicit density in theory becomes a non-issue. In practice, we found Wang & Wang (2007) to work well together with out method, see Appendix C.1.

## 5 EXPERIMENTS

### 5.1 UNIVARIATE REGRESSION ON SYNTHETIC TASKS

In theory, both MDN and our new CRPS-based networks can learn to represent arbitrary univariate distributions. To test their expressiveness visually, we train both methods to learn a synthetic

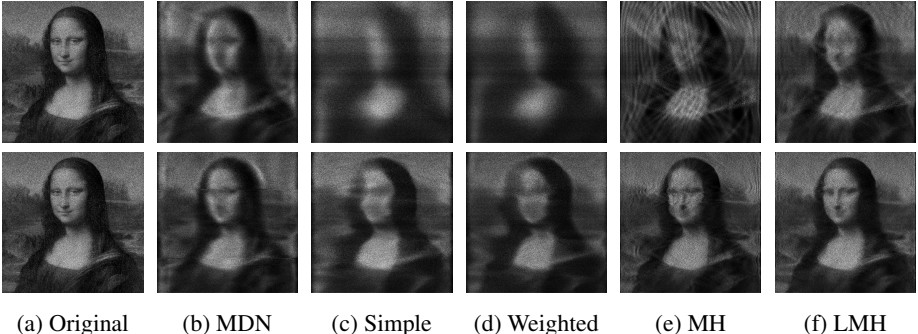

(a) Original     (b) MDN     (c) Simple     (d) Weighted     (e) MH     (f) LMH

Figure 2: Learning the Mona Lisa as output distribution after 40 epochs (top) and after 2000 epochs (bottom). Each row of pixels in the images displays the heatmap of a predicted density based on the row number as input. MH stands for *multiple heads*, while LMH stands for *layered multiple heads*.

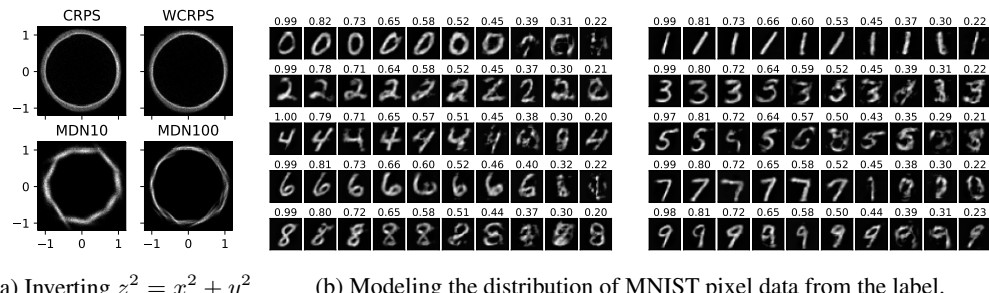

(a) Inverting $z^2 = x^2 + y^2$        (b) Modeling the distribution of MNIST pixel data from the label.

Figure 3: Multivariate tasks. Image (a) shows 2D densities, while (b) displays random samples.

distribution $x \rightarrow p(y)$, where we sample $(x, y)$ points based on the luminance of an image of the Mona Lisa with $x$ as row, and $y$ as column index. As a result, the methods are required to learn many different, but continuously changing distributions, for each row of the image. Note that during training, the models only ever see separate $(x, y)$ pairs at a time, i.e., not $(x; y_1, \ldots, y_n)$ samples.

We train multiple models of the same size, including an MDN model with 100 components, along with multiple kinds of CRPS-based models, and plot the output distributions as 2D heatmaps on Figure 2 after training for 40 and 2000 epochs. The MDN model learns quickly, however ends up with an overly smooth representation. Both single-head CRPS models fail to ultimately represent the details of the image, with perhaps the weighted version performing slightly better. The multi-head and layered-multi-head model both learn fast and manage to learn the baseline distribution in high-detail. However, without layering, visible artifacts remain even after 2000 epochs. Ultimately, the layered-multi-head variant learns fast and ends up representing the distribution with high-accuracy at the same time. See Appendix C.3 for experimental setup and more samples during training.

## 5.2 MULTIVARIATE REGRESSION

For multivariate regression, we run two experiments, one with synthetic and one with real-world data. For exact experimental setups, please refer to Appendix C.4, or the code repository. First, we train the networks to produce solutions to the equation $z^2 = x^2 + y^2$ with $z$ as input and the joint distribution of $(x, y)$ as output. We plot the resulting distributions for $z = 1$ in Figure 3a for four models of the same size: a simple (CRPS) and a weighted (WCRPS) CRPS-based model, and MDN models with 10 (MDN10) and 100 components (MDN100). As we can observe, all models learn to invert the equation, however the CRPS-based models provide a much more robust contour of the resulting circle-shaped distribution.

Having successfully learned the joint distribution of 2 variables, we now turn to a much more challenging task: learning the joint distribution of 784 variables. We reverse the input-output structure of the MNIST classification task (LeCun, 1998), turning it into a multivariate regression problem, with

the label as input and image pixels as output, similar to generative modeling, however without the use of locality structures. Note that we model MNIST solely to stress-test the multivariate formula, not for actual generative modeling purposes. In Figure 3b, we display samples from the predicted distributions. Since multivariate likelihood is hard to estimate, we instead indicate the likelihood of the sampled latent noise corresponding to the sampled outputs. Remarkably, the model is able to learn the joint distribution of the 784 output variables, producing coherent hand-written digits.

## 5.3 UCI REGRESSION UNCERTAINTY BENCHMARK AND BASELINES

The UCI Benchmark (Hernández-Lobato & Adams, 2015) has become the standard benchmark to measure regression uncertainty, with many papers reporting RMSE and (negative) log-likelihood (NLL) results along with standard error. We run our experiments on the exact train-test splits defined by Hernández-Lobato & Adams (2015), using the setup and code from the repository of Gal & Ghahramani (2016). See Appendix C.5 for discussion of hyperparameters and optimization.

Strongest baselines are Sun et al. (2017) (**PBP-MV**) for BNN, and Gal & Ghahramani (2016) (**Dropout**), covering best overall NLL from the literature (Mukhoti et al., 2018), except for boston (**2.33**, El-Laham et al. (2023)), naval ($-5.64$, Lakshminarayanan et al. (2017)) and yacht (**1.10**, Springenberg et al. (2016)). We run our experiments with the network sizes of Sun et al. (2017) (PBP-MV), using two hidden layers with 50 neurons each. We reimplement the method of Lakshminarayanan et al. (2017) (**Ensembles**) and evaluate it at the same network size. For **MDN**, we use our own robust implementation, achieving much stronger scores then previously reported, see Appendix C.2. Baselines compute NLL analytically, while CRPS-based methods are evaluated as described in Section 4.7. For more discussion of baselines, please refer to Appendix C.6. Further comparison against simpler models is presented in Appendix C.11. We compare against Bouchacourt et al. (2016) in Appendix C.13 and against Depeweg et al. (2018) in Appendix C.14.

### 5.3.1 RESULTS

We benchmark multiple variants of our CRPS-based networks, including versions with unweighted, weighted, multi-head, layered-multi-head versions, with and without ensembling. The overall best-performing variant turns out to be the single-head weighted version with ensembling (WCRPS$_e$) on Tables 1 and 2, where we present negative log-likelihood (NLL) and root mean square error (RMSE) scores along with baseline results. Scores without ensembling are also included (WCRPS). The most direct competition for WCRPS$_e$ and WCRPS are MDN$_{bnn}$ and MDN respectively, i.e., non-Gaussian methods with and without techniques for epistemic uncertainty. Best scores are highlighted in **bold**, however one should also note the error ranges included with scores when reading the results.

Overall, it is clear that relaxing the Gaussian assumption generally helps achieve higher NLL scores, except on *concrete* with Dropout winning, and *naval* where a simple BNN dominates ($-6.84$, see Appendix C.11). Strongest scores are mostly shared between MDN$_{bnn}$ and WCRPS$_e$, with the latter scoring best overall. We present scores for variants of CRPS-based methods in an ablation study in Appendix C.7, revealing that the method is also capable of achieving the best NLL result on the *concrete* dataset, with a score of **2.73** (MH variant with ensembling). Multi-head variants underperform WCRPS$_e$ on many (though not all) datasets, indicating that the single-head architecture's strong implicit bias towards smooth distributions appears to be an advantage in most cases.

The *wine* dataset is unique as it is in fact a classification task with 6 discrete target values (see Appendix C.10). MDN outperforms others by 5 orders of magnitude in likelihood by using separate mixture components with increasingly small variances on each value. Note that this approach can theoretically produce NLL values converging to $-\infty$ even while assigning equal probabilities to all outputs, thus offering no useful prediction. Hence, on discrete datasets, NLL is an unreliable metric.

In RMSE, WCRPS$_e$ dominates, with only a small advantage of MDN$_{bnn}$ on *yacht*. As an explanation for the superior performance, while CRPS includes the mean absolute error (MAE) explicitly, all other methods optimize for NLL, which only indirectly optimizes the expectation of the prediction.

Total running time depends not only on raw numeric computation, but also on factors like batch size, data access patterns, convergence speed, and evaluation speed for early stopping. To provide a sense of relative speed, we report running time on the full UCI benchmark suite, which includes datasets with varying characteristics. With MDN at just a bit over 1.5 hours, WCRPS completes in about

Table 1: NLL results on the UCI Datasets benchmark (lower is better)

| dataset | n | Dropout | PBP-MV | Ensembles | MDN | MDN$_{bnn}$ | WCRPS | WCRPS$_e$ |
|---|---|---|---|---|---|---|---|---|
| boston | 0.5k | 2.40$_{\pm0.04}$ | 2.54$_{\pm0.08}$ | 2.44$_{\pm0.05}$ | 2.52$_{\pm0.04}$ | 2.36$_{\pm0.03}$ | 2.40$_{\pm0.05}$ | **2.32**$_{\pm0.05}$ |
| concrete | 1.0k | **2.94**$_{\pm0.02}$ | 3.04$_{\pm0.03}$ | 2.97$_{\pm0.04}$ | 3.13$_{\pm0.04}$ | 2.99$_{\pm0.03}$ | 3.09$_{\pm0.04}$ | 2.95$_{\pm0.05}$ |
| energy | 0.8k | 1.21$_{\pm0.01}$ | 1.01$_{\pm0.01}$ | 0.62$_{\pm0.08}$ | 0.77$_{\pm0.04}$ | 0.65$_{\pm0.02}$ | 0.47$_{\pm0.04}$ | **0.31**$_{\pm0.03}$ |
| kin8nm | 8.2k | -1.14$_{\pm0.01}$ | -1.28$_{\pm0.01}$ | -1.34$_{\pm0.00}$ | -1.22$_{\pm0.01}$ | -1.30$_{\pm0.01}$ | -1.32$_{\pm0.01}$ | **-1.38**$_{\pm0.01}$ |
| naval | 11.9k | -4.45$_{\pm0.00}$ | -4.85$_{\pm0.06}$ | -6.49$_{\pm0.02}$ | -6.24$_{\pm0.04}$ | **-6.72**$_{\pm0.01}$ | -6.65$_{\pm0.05}$ | -6.51$_{\pm0.03}$ |
| power | 9.6k | 2.81$_{\pm0.01}$ | 2.78$_{\pm0.01}$ | 2.69$_{\pm0.01}$ | 2.65$_{\pm0.01}$ | 2.62$_{\pm0.01}$ | 2.66$_{\pm0.01}$ | **2.57**$_{\pm0.01}$ |
| protein | 45.7k | 2.87$_{\pm0.00}$ | 2.77$_{\pm0.01}$ | 2.66$_{\pm0.02}$ | 2.00$_{\pm0.01}$ | **1.96**$_{\pm0.01}$ | 2.22$_{\pm0.01}$ | 2.05$_{\pm0.01}$ |
| wine | 1.6k | 0.93$_{\pm0.01}$ | 0.97$_{\pm0.01}$ | 0.91$_{\pm0.01}$ | -3.48$_{\pm0.03}$ | **-5.11**$_{\pm0.09}$ | 0.90$_{\pm0.03}$ | 0.80$_{\pm0.03}$ |
| yacht | 0.3k | 1.25$_{\pm0.02}$ | 1.64$_{\pm0.02}$ | -0.02$_{\pm0.04}$ | 0.26$_{\pm0.09}$ | 0.63$_{\pm0.04}$ | 0.07$_{\pm0.06}$ | **-0.16**$_{\pm0.05}$ |

Table 2: RMSE results on the UCI Datasets benchmark (lower is better)

| dataset | n | Dropout | PBP-MV | Ensembles | MDN | MDN$_{bnn}$ | WCRPS | WCRPS$_e$ |
|---|---|---|---|---|---|---|---|---|
| boston | 0.5k | 3.61$_{\pm0.23}$ | 3.11$_{\pm0.15}$ | 3.37$_{\pm0.17}$ | 3.73$_{\pm0.25}$ | 2.93$_{\pm0.20}$ | 3.07$_{\pm0.23}$ | **2.91**$_{\pm0.18}$ |
| concrete | 1.0k | 5.45$_{\pm0.19}$ | 5.08$_{\pm0.14}$ | 5.19$_{\pm0.19}$ | 5.99$_{\pm0.12}$ | 5.25$_{\pm0.11}$ | 5.51$_{\pm0.14}$ | **4.94**$_{\pm0.17}$ |
| energy | 0.8k | 0.97$_{\pm0.06}$ | 0.45$_{\pm0.01}$ | 0.86$_{\pm0.12}$ | 1.17$_{\pm0.09}$ | 0.51$_{\pm0.01}$ | 0.45$_{\pm0.02}$ | **0.41**$_{\pm0.01}$ |
| kin8nm | 8.2k | 0.09$_{\pm0.00}$ | 0.07$_{\pm0.00}$ | 0.07$_{\pm0.00}$ | 0.07$_{\pm0.00}$ | 0.07$_{\pm0.00}$ | 0.07$_{\pm0.00}$ | **0.06**$_{\pm0.00}$ |
| naval[2] | 11.9k | 0.00$_{\pm0.00}$ | 0.00$_{\pm0.00}$ | 0.00$_{\pm0.00}$ | 0.00$_{\pm0.00}$ | 0.00$_{\pm0.00}$ | 0.00$_{\pm0.00}$ | 0.00$_{\pm0.00}$ |
| power | 9.6k | 4.18$_{\pm0.04}$ | 3.91$_{\pm0.04}$ | 3.73$_{\pm0.03}$ | 3.93$_{\pm0.04}$ | 3.86$_{\pm0.04}$ | 3.79$_{\pm0.04}$ | **3.62**$_{\pm0.04}$ |
| protein | 45.7k | 4.39$_{\pm0.02}$ | 3.94$_{\pm0.02}$ | 4.20$_{\pm0.03}$ | 3.96$_{\pm0.02}$ | 4.09$_{\pm0.02}$ | 3.76$_{\pm0.02}$ | **3.47**$_{\pm0.02}$ |
| wine | 1.6k | 0.66$_{\pm0.01}$ | 0.64$_{\pm0.01}$ | **0.63**$_{\pm0.00}$ | 0.66$_{\pm0.01}$ | 0.66$_{\pm0.01}$ | 0.64$_{\pm0.01}$ | **0.63**$_{\pm0.01}$ |
| yacht | 0.3k | 1.23$_{\pm0.37}$ | 0.81$_{\pm0.06}$ | 0.92$_{\pm0.08}$ | 1.97$_{\pm0.25}$ | **0.77**$_{\pm0.07}$ | 0.96$_{\pm0.13}$ | 0.78$_{\pm0.09}$ |

2.1$\times$, Ensembles in 3.4$\times$, WCRPS$_{ens}$ in 7$\times$, and MDN$_{bnn}$ in 17.1$\times$ time, the latter being impacted by both slower per-epoch computation and slow convergence. See Appendix C.8 for details.

### 5.3.2 CALIBRATION

Calibration is measured as described in Kuleshov et al. (2018). We take 1000 samples from the predicted distribution, take every 10th as an estimate of percentiles, and measure the ratio of test samples for which the ground truth falls below them. See Figure 4, with the ideal calibration as a dashed red line.

The method displays good calibration on most datasets, except yacht and naval, where it is underconfident. However, on both datasets, the underconfidence is also reflected on the train set (see Appendix C.9), implying that it is most likely a shortcoming of training, e.g., too large learning rate.

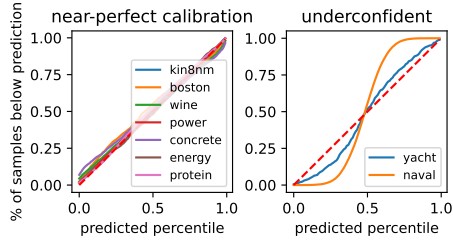

Figure 4: Calibration of WCRPS$_e$ on the UCI Datasets benchmark.

## 6 CONCLUSIONS

In this paper, we proposed optimizing nondeterministic neural networks using loss functions derived from the continuous ranked probability score (CRPS). We derived unbiased $\mathcal{O}(n \log n)$ formulas for both unweighted and weighted sample-based approximations of the CRPS, and proposed multiple architectures for sampling from the aleatoric distribution of the target, enabling truly distribution-free learning. Through extensive experiments on both univariate and multivariate synthetic and real-world datasets, our method consistently demonstrates superior performance compared to baseline models, making it an attractive choice for capturing aleatoric uncertainty. Ultimately, we believe CRPS optimization is an effective and novel way of training neural networks for predicting regression uncertainty in a distribution-free manner, while at the same time remaining conceptually simple and easy to implement. Finally, we make our implementations and experiment code freely available.

---

[2]We have no exact baseline RMSE scores for *naval*, as prior work reports only 2 decimals (Sun et al., 2017).

REPRODUCIBILITY

We make all code and hyperparameters used to run the experiments in the paper publicly available in our source code repository at `https://github.com/proto-n/torch-naut`.

ACKNOWLEDGMENTS

Supported by the European Union project RRF-2.3.1-21-2022-00004 within the framework of the Artificial Intelligence National Laboratory Program and and by the European Defence Fund project 101103386 *FaRADAI*.

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

# A    BACKGROUND

## A.1    LIST OF PRIOR WORK EVALUATING ON THE UCI REGRESSION UNCERTAINTY BENCHMARK

The UCI Benchmark was originally defined by Hernández-Lobato & Adams (2015). We are aware of results being reported for the benchmark in Louizos & Welling (2016); Springenberg et al. (2016); Gal & Ghahramani (2016); Sun et al. (2017); Lakshminarayanan et al. (2017); Mukhoti et al. (2018); Ghosh et al. (2019); Amini et al. (2020); Goulet et al. (2021); El-Laham et al. (2023); Gawlikowski et al. (2023); Deka et al. (2024).

## A.2    SMALL SURVEY OF UNCERTAINTY SURVEYS

To provide evidence for our claim that the distribution-free modeling of aleatoric uncertainty for neural network regression is a relatively under-researched and under-reported area, we conduct a small survey, where we list uncertainty-adjacent survey papers and examine them based on four criteria:

MDN   Whether the survey mentions the use of mixture models for aleatoric uncertainty in regression, cites MDN (Bishop, 1994), or other works using MDN

NF   Whether the survey mentions the use of Normalizing Flows for aleatoric uncertainty

MMD   Whether the survey mentions the Maximum Mean Discrepancy loss in the context of learning aleatoric uncertainty

ES   Whether the survey mentions the Energy score / CRPS in the context of aleatoric uncertainty or cites DISCO Nets (Bouchacourt et al., 2016)

We summarize our findings in Table 3. To elaborate further on each:

- Goan & Fookes (2020) is a survey of Bayesian Neural Networks, and as such is primarily concerned with epistemic uncertainty. However, it does cite SiriTeam (2017) which uses MDN.

- Abdar et al. (2021) is a general survey of uncertainty techniques in deep learning. In the main text, it describes MDN under the name *Gaussian Mixture Model* and cites Choi et al. (2018) which uses MDN.

- Kabir et al. (2018) is survey of uncertainty quantification in deep learning. Does not mention or cite any of the above.

- Hüllermeier & Waegeman (2021) is a general introduction to epistemic and aleatoric uncertainty in machine learning. Does not mention or cite any of the above.

- Jospin et al. (2022) is a tutorial on Bayesian Neural Networks, and (somewhat appropriately in this case) does not mention or cite any of the above

- Magris & Iosifidis (2023) is a survey on Bayesian learning for neural networks in general. Does not mention or cite any of the above.

- Gawlikowski et al. (2023) is a survey of uncertainty in deep neural networks. Cites Corduneanu & Bishop (2001), therefore indirectly references MDN.

- Tyralis & Papacharalampous (2024) is a review of predictive uncertainty in machine learning. It does describe the CRPS, but mentions uses for learning only in context of ensembles or with a Gaussian assumption. It also mentions the Energy Score, but does not describe any uses. Does not cite Bouchacourt et al. (2016).

Many of these works describe using non-Gaussian distributions, however usually in the context of variational distributions, (i.e., the distribution of weights in a BNN). Some mention the use of MMD for quantile regression or calibration, but not for distributional regression.

Table 3: A small survey of uncertainty surveys.

| Paper | MDN | NF | MMD | ES |
|---|---|---|---|---|
| Goan & Fookes (2020) | ✓ | - | - | - |
| Abdar et al. (2021) | ✓ | ✓ | - | - |
| Kabir et al. (2018) | - | - | - | - |
| Hüllermeier & Waegeman (2021) | - | - | - | - |
| Jospin et al. (2022) | - | - | - | - |
| Magris & Iosifidis (2023) | - | - | - | - |
| Gawlikowski et al. (2023) | ✓ | - | - | - |
| Tyralis & Papacharalampous (2024) | - | - | - | ✓ |

## B  PROOFS AND DERIVATIONS

### B.1  WEIGHTED CRPS FORMULA

The classic formula estimates the first term of Equation (4) by taking a samples $(\hat{y}_i)$ from $\hat{Y} \sim Q$ and calculating

$$\mathrm{E}\left|y - \hat{Y}_1\right| = \int_\Omega |y - \hat{y}| \, \mathrm{d}Q(\hat{y}) \approx \frac{1}{n} \sum_{i=1}^n |y - \hat{y}_i|. \tag{10}$$

If $Q$ is absolutely continuous w.r.t. $Q'$ (or $Q \ll Q'$), then given $Y' \sim Q'$ and its sample $(y_i')$,

$$\int_\Omega |y - \hat{y}| \, \mathrm{d}Q(\hat{y}) = \int_\Omega |y - \hat{y}| \tfrac{\mathrm{d}Q}{\mathrm{d}Q'}(\hat{y}) \, \mathrm{d}Q'(\hat{y}) = \mathrm{E}[|y - Y'|w(Y')] \approx \frac{1}{n} \sum_{i=1}^n |y - y_i'|w(y_i'), \tag{11}$$

where $w(y') = \frac{\mathrm{d}Q}{\mathrm{d}Q'}(y')$. Doing the same for the second term:

$$\mathrm{E}\left|\hat{Y}_1 - \hat{Y}_2\right| = \int_\Omega \int_\Omega |\hat{y}_1 - \hat{y}_2| \, \mathrm{d}Q(\hat{y}_1) \, \mathrm{d}Q(\hat{y}_2) \tag{12}$$

$$= \int_\Omega \int_\Omega |\hat{y}_1 - \hat{y}_2| \tfrac{\mathrm{d}Q}{\mathrm{d}Q'}(\hat{y}_1) \, \mathrm{d}Q'(\hat{y}_1) \tfrac{\mathrm{d}Q}{\mathrm{d}Q'}(\hat{y}_2) \, \mathrm{d}Q'(\hat{y}_2) \tag{13}$$

$$= \mathrm{E}[|Y_1' - Y_2'|w(Y_1')w(Y_2')] \approx \frac{1}{n^2} \sum_{i=1}^n \sum_{j=1}^n |y_i' - y_j'|w(y_i')w(y_j'), \tag{14}$$

with $Y_1', Y_2' \sim Q'$ iid. Finally,

$$\mathrm{CRPS}(Q, y) \approx \mathrm{CRPS}_w(F_{\mathrm{ECDF}}^w, y) = \frac{1}{n} \sum_{i=1}^n w_i |y - \hat{y}_i| - \frac{1}{2n^2} \sum_{i=1}^n \sum_{j=1}^n w_i w_j |\hat{y}_i - \hat{y}_j|, \tag{15}$$

where $w_i$ further denotes $w(\hat{y}_i)$, and $F_{\mathrm{ECDF}}^w$ denotes the empirical cumulative distribution function of the weighted sample $(\hat{y}_i, w_i)$. An unbiased version can be computed as in Section 4.1 by replacing the $\frac{1}{2n^2}$ coefficient by $\frac{1}{2n(n-1)}$.

### B.2  FAST BIASED VERSION

First we are going to prove a formula for the biased version for the sake of simplicity, and describe how to get an unbiased version in Appendix B.3. The following formula can be calculated in $\mathcal{O}(n \log n)$ time by first ordering the sample to get $\hat{y}_{(1)}, \ldots, \hat{y}_{(n)}$ and corresponding weights $w_{(1)}, \ldots, w_{(n)}$.

**Proposition B.1.** *Let $W = \sum_{i=1}^n w_{(i)}$ and $s_i = \sum_{j=1}^i w_{(j)}$. Then*

$$\mathrm{CRPS}_w(F_{ECDF}^w, y) = \frac{2}{n^2} \sum_{i=1}^n w_{(i)}(\hat{y}_{(i)} - y)\left(n\mathbb{1}\{y < \hat{y}_{(i)}\} - s_i + \frac{W - n + w_{(i)}}{2}\right) \tag{16}$$

*Proof.* Using counting method. First observe that

$$\frac{1}{n}\sum_{i=1}^{n}w_{(i)}|y-\hat{y}_{(i)}| - \frac{1}{2n^2}\sum_{i=1}^{n}\sum_{j=1}^{n}w_{(i)}w_{(j)}|\hat{y}_{(i)}-\hat{y}_{(j)}| \tag{17}$$

$$= -\frac{1}{n^2}\left(\sum_{i=1}^{n}\sum_{j=1}^{n}\mathbb{1}\{i<j\}\,w_{(i)}w_{(j)}|\hat{y}_{(i)}-\hat{y}_{(j)}| - n\sum_{i=1}^{n}w_{(i)}|y-\hat{y}_{(i)}|\right) \tag{18}$$

Next, we are going to count the $|\hat{y}_{(i)} - y|$ distances on Figure 5, where we visually display the summands and their weights from Equation (18) (omitting the preceding $(-\frac{1}{n^2})$ coefficient)

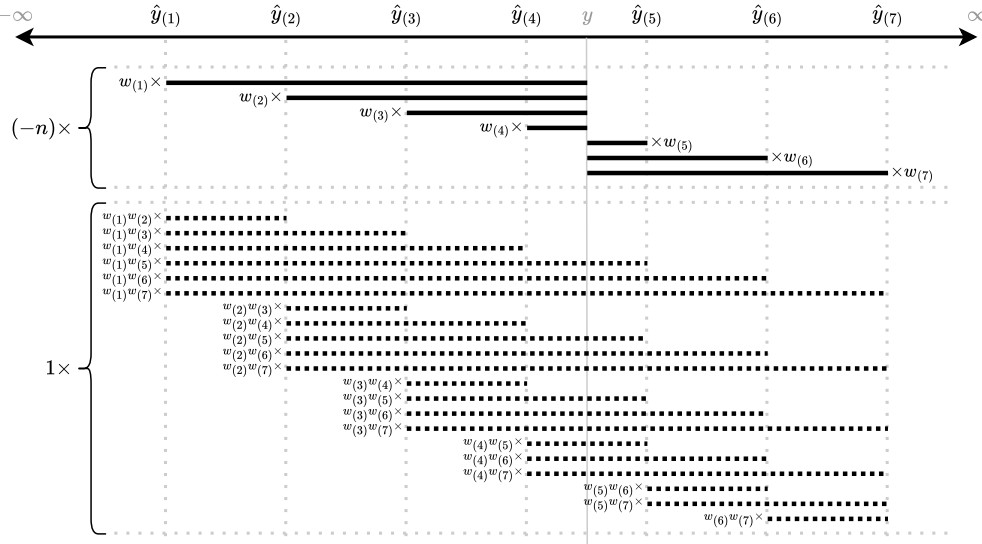

Figure 5: Counting method example with $n = 7$. We are counting the sum of coefficients for each $|\hat{y}_{(i)} - y|$ term. On the figure, solid lines are to be counted with a coefficient of $(-n)$ and dotted lines with a coefficient of $1$. Further, individual change-of-measure coefficients are indicated next to each line in the form $w_{(i)}$ or $w_{(i)}w_{(j)}$. The global coefficient $(-\frac{1}{n^2})$ is omitted for simplicity.

Starting from the left with the example of $i = 1$, for $(y - \hat{y}_{(1)})$, we have the coefficients

$$-nw_{(1)} + w_{(1)}\sum_{j=2}^{n}w_{(j)}. \tag{19}$$

However, this way we over-count each $(y - \hat{y}_{(j)})$ term for $\{j > i,\ \hat{y}_{(j)} < y\}$ by a factor of $w_{(1)}w_{(j)}$, which we are going to correct for when counting the coefficients for $(y - \hat{y}_{(j)})$.

Accordingly, for the $i$-th term $(y - \hat{y}_{(i)})$ in general, assuming $\hat{y}_{(i)} < y$, we have the coefficients

$$-nw_{(i)} + w_{(i)}\sum_{j=i+1}^{n}w_{(j)} - w_{(i)}\sum_{j=1}^{i-1}w_{(j)} \tag{20}$$

$$= w_{(i)}\left(-n - \sum_{j=1}^{i-1}w_{(j)} + \left(W - \sum_{j=1}^{i}w_{(j)}\right)\right) \tag{21}$$

$$= w_{(i)}\left(W - n - 2s_i + w_{(i)}\right). \tag{22}$$

We can proceed to count the coefficients of $(\hat{y}_{(i)} - y)$ with $y < \hat{y}_{(i)}$ by mirroring the above logic. Denoting $k_i = W - \sum_{j=1}^{i-1} w_{(j)} = W - s_i + w_{(i)}$, we have the coefficients (cf. Equation (22))

$$w_i\left(W - n - 2k_i + w_{(i)}\right) = w_{(i)}\left(W - n - 2(W - s_i + w_{(i)})\right) + w_{(i)}\right) \tag{23}$$

$$= -w_{(i)}(W + n - 2s_i + w_{(i)}). \tag{24}$$

Finally, we have

$$\mathrm{CRPS}_w(F_{\mathrm{ECDF}}^w, y) \tag{25}$$

$$= -\frac{1}{n^2}\left(\sum_{\hat{y}_{(i)}<y} w_{(i)}\left(W - n - 2s_i + w_{(i)}\right)(y - \hat{y}_{(i)}) - \sum_{y<\hat{y}_{(i)}} w_{(i)}\left(W + n - 2s_i + w_{(i)}\right)(\hat{y}_{(i)} - y)\right) \tag{26}$$

$$= \frac{2}{n^2}\sum_{i=1}^{n} w_{(i)}(\hat{y}_{(i)} - y)\left(n\mathbb{1}\{y < \hat{y}_{(i)}\} - s_i + \frac{W - n + w_{(i)}}{2}\right). \tag{27}$$

$\square$

### B.3 Fast unbiased version

**Proposition B.2.** *An unbiased version of Equation* (16) *can also be calculated in* $\mathcal{O}(n\log n)$.

*Proof.*

$$\mathrm{CRPS}_w(F_{\mathrm{ECDF}}^w, y) = \frac{2}{n^2}\sum_{i=1}^{n} w_{(i)}(\hat{y}_{(i)} - y)\left(n\mathbb{1}\{y < \hat{y}_{(i)}\} - s_i + \frac{W - n + w_{(i)}}{2}\right) \tag{28}$$

$$= \frac{1}{n^2}\left(\sum_{y<\hat{y}_{(i)}} w_{(i)}\left(\hat{y}_{(i)} - y\right)\left(n - 2s_i + W + w_{(i)}\right) + \sum_{\hat{y}_{(i)}<y} w_{(i)}\left(y - \hat{y}_{(i)}\right)\left(n + 2s_i - W - w_{(i)}\right)\right) \tag{29}$$

Notice that

$$\frac{1}{n}\sum_{i=1}^{n} w_{(i)}|y - \hat{y}_{(i)}| = \frac{1}{n^2}\left(\sum_{y<\hat{y}_{(i)}} w_{(i)}\left(\hat{y}_{(i)} - y\right)n + \sum_{\hat{y}_{(i)}<y} w_{(i)}\left(y - \hat{y}_{(i)}\right)n\right), \tag{30}$$

and that Equation (30) appears in Equation (29). Therefore, we can modify Equation (29) to express the unbiased coefficient for the second term of Equation (7) without affecting the first term by taking

$$\frac{1}{n(n-1)}\left(\sum_{y<\hat{y}_{(i)}} w_{(i)}\left(\hat{y}_{(i)} - y\right)\left((n-1) - 2s_i + W + w_{(i)}\right) + \tag{31}$$

$$+ \sum_{\hat{y}_{(i)}<y} w_{(i)}\left(y - \hat{y}_{(i)}\right)\left((n-1) + 2s_i - W - w_{(i)}\right)\right) \tag{32}$$

$$= \frac{2}{n(n-1)}\left(\sum_{y<\hat{y}_{(i)}} w_{(i)}\left(\hat{y}_{(i)} - y\right)\left(n - 1 - s_i + \frac{W - n + w_{(i)} + 1}{2}\right) + \tag{33}$$

$$+ \sum_{\hat{y}_{(i)}<y} w_{(i)}\left(\hat{y}_{(i)} - y\right)\left(-s_i + \frac{W - n + w_{(i)} + 1}{2}\right)\right) \tag{34}$$

Transforming back to short form and denoting the unbiased version with $\mathrm{CRPS}'_w$, we get

$$\mathrm{CRPS}'_w(F_{\mathrm{ECDF}}^w, y) = \frac{2}{n(n-1)}\sum_{i=1}^{n} w_{(i)}\left(\hat{y}_{(i)} - y\right)\left((n-1)\mathbb{1}\{y < \hat{y}_{(i)}\} - s_i + \frac{W - n + w_{(i)} + 1}{2}\right). \tag{35}$$

$\square$

### B.4 FAST UNBIASED UNWEIGHTED VERSION

**Proposition B.3.**

$$\text{CRPS}'(F_{ECDF}, y) = \frac{2}{n(n-1)} \sum_{i=1}^{n} \left( \tilde{y}_{(i)} - y \right) \left( (n-1) \mathbb{1}\{y < \tilde{y}_{(i)}\} - i + 1 \right), \qquad (36)$$

*Proof.* Taking $w_{(i)} \equiv 1$, we get $W = n$ and $s_i = i$. Substituting into Equation (35), the proposition immediately follows. $\qquad\square$

## C IMPLEMENTATION DETAILS AND RESULTS

### C.1 KERNEL DENSITY ESTIMATION

As described, we use the adaptive-bandwidth kernel density estimator of Wang & Wang (2007) for likelihood-estimation in our experiments. We implement a GPU-based version of the algorithm, based on the C++ based cpu-implementation of the awKDE[3] Python library.

The method works by first creating a naive pilot likelihood estimate in each sampled point based on the full sample using a traditional constant bandwidth KDE, and then setting the local bandwith to be inversely proportional to the pilot-estimate. Calculating the pilot estimates requires calculating the pairwise distance matrix for all sampled points, making the algorithm quadratic in the number of points. Since calculating pairwise distances on a GPU is fast, the quadratic runtime proved to be feasible in our experiments. We run the algorithm with the default parameters for the original implementation of awKDE, i.e., using Silverman's estimate for the global bandwidth parameter (Silverman, 2018).

As Wang & Wang (2007) note, the quality of the pilot estimate is ultimately not critical for the quality of the final estimate. Accordingly, we have seen some success with using a fixed number of samples to produce the pilot-estimate, making the runtime linear instead of quadratic. However, proper investigation of such methods is out of scope for this work.

### C.2 MDN IMPLEMENTATION

MDN models seem to be relatively under-utilized in uncertainty-related literature, with some surveys barely mentioning the method (Abdar et al., 2021; Goan & Fookes, 2020), and others omitting any mention at all (Gawlikowski et al., 2023; Jospin et al., 2022; Magris & Iosifidis, 2023; Kabir et al., 2018; Hüllermeier & Waegeman, 2021; Tyralis & Papacharalampous, 2024). Further, it appears that MDN has very rarely been evaluated as baseline in prior work. We are aware of a single instance where it is used as a baseline method (El-Laham et al., 2023) for the UCI benchmark, however with the authors reporting substantially lower scores than our experiments.

MDN is sometimes also referred to simply as Gaussian mixture model (GMM) (Hubschneider et al., 2019; Abdar et al., 2021), however in general these two terms do not mean the same thing, as GMM includes a larger class of methods with various modeling and optimization strategies. It is also important to note that not every Gaussian mixture model is made equal: in order to approximate even simple asymmetric densities, a high number of Gaussians might be required. Using methods relying on, e.g., ensembling (Jospin et al., 2022), prohibits the use of a high number of mixture components due to computational constraints.

The reason for its neglect is perhaps the fact that MDN appears to be notoriously challenging to implement, often suffering from numerical instability, failure to converge, and mode collapse (Makansi et al., 2019; Brando, 2017; Rupprecht et al., 2017; Cui et al., 2019; Curro & Raquet, 2018; Messaoud et al., 2018; Graves, 2013; Hjorth & Nabney, 1999). Nevertheless, we do implement and evaluate a robust variant of MDN, capable of optimizing the mixture of 100 Gaussian components in our experiments.

---

[3] https://github.com/mennthor/awkde

Our implementation takes a number of steps to avoid numerical instabilities and convergence issues that tend to plague implementations of MDN, this way achieving much stronger scores on the UCI benchmark then previously reported. We apply the following techniques:

- We standardize both the input variables $X$ and the target $Y$ to have 0 mean and a variance of 1.

- We apply a linear learning rate warm-up schedule, where the learning-rate is set to 0.1 of its value at the beginning and increases over 10 epochs.

- We apply gradient clipping at 10 before taking a step along the loss gradient.

- We calculate the log-likelihood in the log-domain as

$$l_i = -\frac{1}{2} \left( \frac{y - \mu_i}{\sigma_i} \right)^2 - \ln \left( \sigma_i \sqrt{2\pi} \right) \tag{37}$$

$$w_i = \pi_i - \mathrm{LSE}(\pi_1, \ldots, \pi_n) \tag{38}$$

$$\text{log-likelihood} = \mathrm{LSE}\left( l_1 + w_1, \ldots, l_n + w_n \right) \tag{39}$$

where LSE denotes the numerically stable implementation of LogSumExp of the PyTorch framework[4], i.e.,

$$\mathrm{LSE}(x_1, \ldots, x_n) = \log \sum_{i=1}^{n} e^{x_i}. \tag{40}$$

- During training, we clamp the log-likelihood loss to over $-20$, as even a single evaluation where the likelihood at $y$ is effectively zero can result in the experiment producing $\infty$ and nan values.

- We clamp the inputs of the $\mathrm{softplus}$ to over $-15$ and the input logits of the $\mathrm{softmax}$ between $-15$ and $15$, as both tend to converge to very low and hight values:
    - as the method turns off certain components, their logits $\pi_i$ tend towards $-\infty$;
    - if the method needs to represent an unimodal distribution, its logit $\pi_i$ tends toward $\infty$;
    - on discrete datasets and in the case of unused components, the logit $\pi_i$ as well as the input of the $\mathrm{softplus}$ $s_i$ tends toward $-\infty$.

See Appendix C.4 for multivariate details.

## C.3  LEARNING THE MONA LISA

For training, we sample 5000 samples for each row from the distribution defined by the luminance of an $500 \times 500$ image of the Mona Lisa, such that the input of the network is the row index, and the expected output is the column index. We train $\mathbb{R} \to \mathbb{R}$ networks with layers of size $[128, 512, 512, 1024]$, GELU activation, SGD optimizer with lr $= 10^{-4}$ and a momentum value of 0.8 for 2000 epochs. See Figure 6 for distributions at 40, 100, 200, 500, 1000 and 2000 epochs. The WCRPS-based method uses 50 heads for both multi-head variants, and 5 layers for the layered-multi-head variant.

## C.4  MULTIVARIATE EXPERIMENTAL SETUP

For the paraboloid task $z^2 = x^2 + y^2$ we generate training data by drawing 2M samples from $X, Y \sim \mathcal{U}(-2, 2)$, filter them to $x^2 + y^2 <= 4$, and calculate the corresponding $z^2$ values. We train $\mathbb{R} \to \mathbb{R}^2$ networks with layers of size $[64, 128, 128]$, GELU activation and LayerNorm, and use an SGD optimizer with 1k epochs and lr $= 10^{-4}$.

We implement a multivariate version of MDN with full covariance matrix modeling based on the Cholesky-decomposition based parameterization of the multivariate normal distribution (Muschinski et al., 2024), while also employing techniques from Appendix C.2 where applicable.

For the MNIST task, we train a network with layers of size $[16, 1024, 2048, 4096]$ with ReLU activation on the train set of MNIST for 1k epochs with lr $= 10^{-5}$ and a momentum value of 0.9

---

[4]https://pytorch.org/docs/stable/generated/torch.logsumexp.html

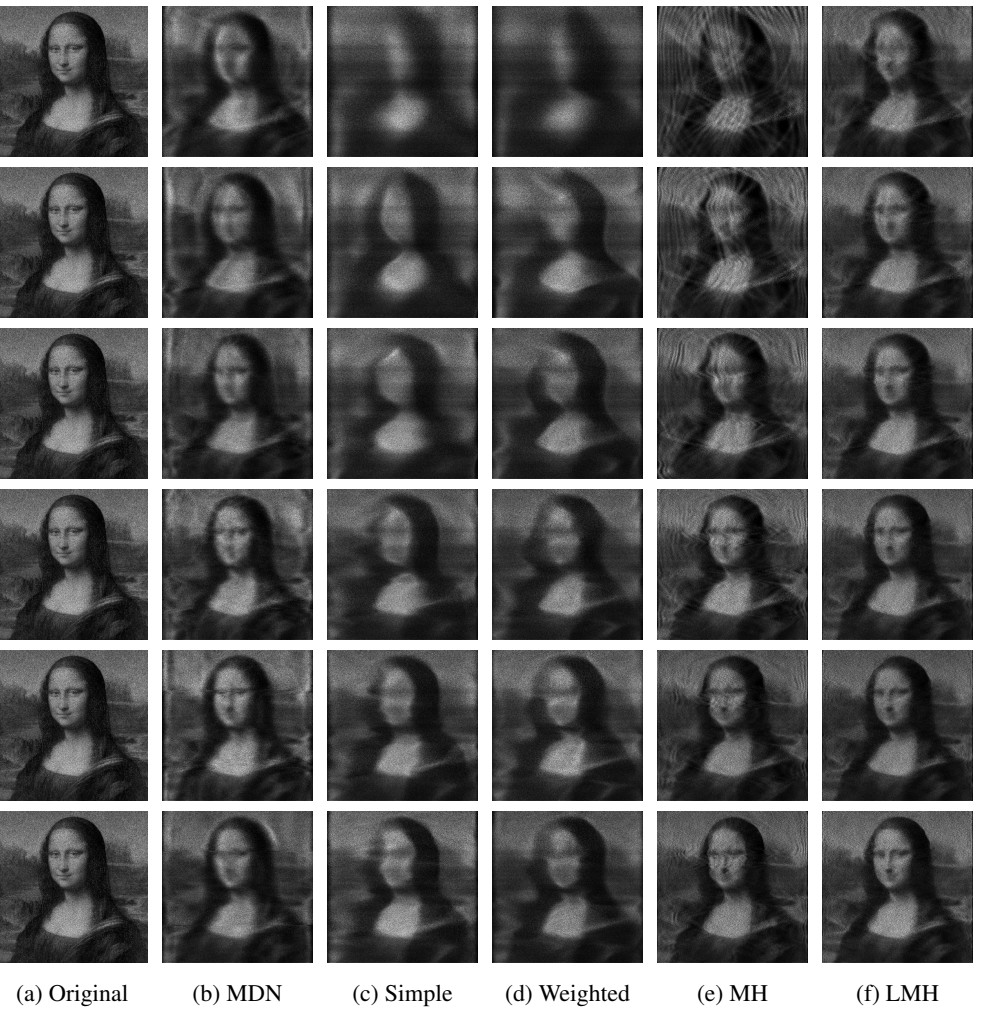

(a) Original     (b) MDN     (c) Simple     (d) Weighted     (e) MH     (f) LMH

Figure 6: Learning the Mona Lisa as output distribution after 40, 100, 200, 500, 1000 and 2000 epochs from top to bottom respectively.

C.5   HYPERPARAMETERS ON THE UCI BENCHMARK

We do not optimize hyperparameters individually for each dataset, however do use adaptive batch sizes, and larger L2 regularization for the 4 smallest datasets. Early stopping is used with 20% of the training set as validation.

In our experiments we use the *AdamW* optimizer with a learning-rate of 0.001 and a linear learning rate warm-up schedule starting with the coefficient 0.1 and ending with 1 after 10 epochs. We use early stopping with a patience value of 50 on validation sets consisting of 20% of the training set for each split. We set L2 regularization over the weights to $10^{-6}$ on all datasets except the four smallest ones (boston, concrete, energy, yacht), where we use $10^{-4}$. Batch size is chosen to be $\max(\sqrt{m} - 5, 1)$ with $m$ being the training set size, for a balance of computational efficiency and predictive accuracy.

We run Ensembles and ensembling-based variants of WCRPS using 5 networks, as in the original paper of Lakshminarayanan et al. (2017). We use 20 heads for all multi-head variants of WCRPS on the UCI benchmark, and 10 layers for all layered-multi-head variants. For evaluating the CRPS formula variants, we take 100 samples during training and 1000 samples for evaluation during early stopping. See Appendix C.12 for a comparison of different choices for the samples drawn during training.

C.6   BASELINES

Most prior work run experiments by using 100 instead of 50 neurons for each hidden layer for the largest (protein) dataset. For simplicity, we use 50 neurons only, as the models seem to perform well regardless.

For our reimplementation of Ensembles, we also employ techniques from Appendix C.2 where applicable. Our results are generally in line with or slightly better than those reported in the original paper at the same network sizes, except for the naval dataset, where our implementation performs significantly better (-5.64 originally, -6.52 in our testing for the single-layer model). Our improvement may be due to one or more enhancements discussed in Appendix C.2, possibly influenced by the unusual data range, as also indicated by the RMSE values in Table 2.

Note, that the standard network size in most prior work is a single layer of 50 neurons. However, the best prior scores in Sun et al. (2017) (PBP-MV) use a network size of two hidden layers with 50 neurons each, which leads us to re-evaluate Ensembles at the same size. Further, we also evaluate MDN and CRPS-based models at the same network size for fairness.

For the CRPS-based models we use GELU (Hendrycks & Gimpel, 2016) activation together with layer normalization (Ba et al., 2016), as it appears to help the network. For correctness and transparency, we also re-evaluate Ensembles, MDN, and MDN$_{bnn}$ with the same setup, and include stronger versions in the main text of the paper in Table 4. See Tables 4 and 5 for pairwise comparisons in NLL and RMSE. According to the results, we include the ReLU versions for Ensembles (5/9 for NLL and 4/7 for RMSE), the ReLU versions for MDN$_{bnn}$ (4/8 for NLL with 3 overall strongest results and 6/8 for RMSE), and the GELU version for MDN (6/9 for NLL and 5/8 for RMSE).

The source-code for the method Dropout is publicly available, however proper evaluation requires grid-search optimization of two separate hyperparameters over every single train-test split of 9 datasets separately, which proved to be prohibitively expensive for the two-layer variant. Therefore, we report the original scores for the single-layer variant in case of Dropout.

Due to methodological differences, we cannot compare against Amini et al. (2020). Amini et al. (2020) propose modeling aleatoric uncertainty using the Normal distribution, and epistemic uncertainty as a parametric meta-distribution over the parameters of the aleatoric normal distribution (Normal-Inverse Gamma). They report good performance on the UCI benchmark. However, we could not include their results: reproducing their experiments and investigating the code provided[5] reveals that

---

[5]`https://github.com/aamini/evidential-deep-learning/tree/main/neurips2020`

Table 4: Pairwise comparison of ReLU and GELU based baseline variants, evaluated in NLL. Pairwise best scores are highlighted with underline. Scores that perform at least on par when compared to the results in Table 1 are highlighted in **bold**.

| dataset | Ensembles | Ensembles$^{\text{gelu}}$ | MDN | MDN$^{\text{gelu}}$ | MDN$_{\text{bnn}}$ | MDN$^{\text{gelu}}_{\text{bnn}}$ | WCRPS$_{\text{e}}$ |
|---|---|---|---|---|---|---|---|
| boston | $2.44_{\pm0.05}$ | $2.41_{\pm0.05}$ | $2.45_{\pm0.06}$ | $2.52_{\pm0.04}$ | $2.36_{\pm0.03}$ | $\mathbf{2.32}_{\pm0.04}$ | $\mathbf{2.32}_{\pm0.05}$ |
| concrete | $2.97_{\pm0.04}$ | $3.14_{\pm0.07}$ | $3.16_{\pm0.03}$ | $3.13_{\pm0.04}$ | $2.99_{\pm0.03}$ | $2.99_{\pm0.03}$ | $2.95_{\pm0.05}$ |
| energy | $0.62_{\pm0.08}$ | $0.75_{\pm0.15}$ | $1.09_{\pm0.06}$ | $0.77_{\pm0.04}$ | $0.65_{\pm0.02}$ | $0.58_{\pm0.03}$ | $\mathbf{0.31}_{\pm0.03}$ |
| kin8nm | $-1.34_{\pm0.00}$ | $-1.37_{\pm0.01}$ | $-1.15_{\pm0.01}$ | $-1.22_{\pm0.01}$ | $-1.30_{\pm0.01}$ | $-1.37_{\pm0.01}$ | $\mathbf{-1.38}_{\pm0.01}$ |
| naval | $-6.49_{\pm0.02}$ | $-6.12_{\pm0.02}$ | $-6.55_{\pm0.03}$ | $-6.24_{\pm0.04}$ | $\mathbf{-6.72}_{\pm0.01}$ | $-6.38_{\pm0.02}$ | $-6.51_{\pm0.03}$ |
| power | $2.69_{\pm0.01}$ | $2.74_{\pm0.02}$ | $2.67_{\pm0.00}$ | $2.65_{\pm0.01}$ | $2.62_{\pm0.01}$ | $2.64_{\pm0.01}$ | $\mathbf{2.57}_{\pm0.01}$ |
| protein | $2.66_{\pm0.02}$ | $2.64_{\pm0.05}$ | $2.02_{\pm0.01}$ | $2.00_{\pm0.01}$ | $\mathbf{1.96}_{\pm0.01}$ | $1.97_{\pm0.01}$ | $2.05_{\pm0.01}$ |
| wine | $0.91_{\pm0.01}$ | $0.96_{\pm0.02}$ | $-3.75_{\pm0.06}$ | $-3.48_{\pm0.03}$ | $\mathbf{-5.11}_{\pm0.09}$ | $-3.91_{\pm0.04}$ | $0.80_{\pm0.03}$ |
| yacht | $-0.02_{\pm0.04}$ | $\mathbf{-0.32}_{\pm0.06}$ | $0.79_{\pm0.09}$ | $0.26_{\pm0.09}$ | $0.63_{\pm0.04}$ | $0.55_{\pm0.03}$ | $\mathbf{-0.16}_{\pm0.05}$ |

Table 5: Pairwise comparison of ReLU and GELU based baseline variants, evaluated in RMSE. Pairwise best scores are highlighted with underline.

| dataset | Ensembles | Ensembles$^{\text{gelu}}$ | MDN | MDN$^{\text{gelu}}$ | MDN$_{\text{bnn}}$ | MDN$^{\text{gelu}}_{\text{bnn}}$ | WCRPS$_{\text{e}}$ |
|---|---|---|---|---|---|---|---|
| boston | $3.37_{\pm0.17}$ | $3.27_{\pm0.24}$ | $3.54_{\pm0.27}$ | $3.73_{\pm0.25}$ | $2.93_{\pm0.20}$ | $3.10_{\pm0.22}$ | $2.91_{\pm0.18}$ |
| concrete | $5.19_{\pm0.19}$ | $5.47_{\pm0.16}$ | $6.34_{\pm0.11}$ | $5.99_{\pm0.12}$ | $5.25_{\pm0.11}$ | $5.36_{\pm0.14}$ | $4.94_{\pm0.17}$ |
| energy | $0.86_{\pm0.12}$ | $0.52_{\pm0.02}$ | $1.89_{\pm0.13}$ | $1.17_{\pm0.09}$ | $0.51_{\pm0.01}$ | $0.61_{\pm0.03}$ | $0.41_{\pm0.01}$ |
| kin8nm | $0.07_{\pm0.00}$ | $0.07_{\pm0.00}$ | $0.08_{\pm0.00}$ | $0.07_{\pm0.00}$ | $0.07_{\pm0.00}$ | $0.06_{\pm0.00}$ | $0.06_{\pm0.00}$ |
| naval | $0.00_{\pm0.00}$ | $0.00_{\pm0.00}$ | $0.00_{\pm0.00}$ | $0.00_{\pm0.00}$ | $0.00_{\pm0.00}$ | $0.00_{\pm0.00}$ | $0.00_{\pm0.00}$ |
| power | $3.73_{\pm0.03}$ | $3.83_{\pm0.04}$ | $3.96_{\pm0.04}$ | $3.93_{\pm0.04}$ | $3.86_{\pm0.04}$ | $3.95_{\pm0.04}$ | $3.62_{\pm0.04}$ |
| protein | $4.20_{\pm0.03}$ | $4.09_{\pm0.03}$ | $4.07_{\pm0.03}$ | $3.96_{\pm0.02}$ | $4.09_{\pm0.02}$ | $4.11_{\pm0.02}$ | $3.47_{\pm0.02}$ |
| wine | $0.63_{\pm0.00}$ | $0.64_{\pm0.01}$ | $0.65_{\pm0.01}$ | $0.66_{\pm0.01}$ | $0.66_{\pm0.01}$ | $0.64_{\pm0.01}$ | $0.63_{\pm0.01}$ |
| yacht | $0.92_{\pm0.08}$ | $1.24_{\pm0.15}$ | $1.68_{\pm0.18}$ | $1.97_{\pm0.25}$ | $0.77_{\pm0.07}$ | $0.83_{\pm0.08}$ | $0.78_{\pm0.09}$ |

Table 6: NLL results on the UCI Datasets benchmark (lower is better). Scores that perform at least on par when compared to the results in Table 1 are highlighted in **bold**.

| dataset | CRPS | WCRPS | WCRPS$^{mh}$ | WCRPS$^{lmh}$ | WCRPS$_e$ | WCRPS$_e^{mh}$ | WCRPS$_e^{lmh}$ |
|---|---|---|---|---|---|---|---|
| boston | $2.38_{\pm 0.06}$ | $2.40_{\pm 0.05}$ | $2.40_{\pm 0.04}$ | $2.40_{\pm 0.06}$ | $\mathbf{2.32}_{\pm 0.05}$ | $\mathbf{2.32}_{\pm 0.05}$ | $\mathbf{2.30}_{\pm 0.04}$ |
| concrete | $3.11_{\pm 0.04}$ | $3.09_{\pm 0.04}$ | $2.97_{\pm 0.04}$ | $\mathbf{2.89}_{\pm 0.03}$ | $2.95_{\pm 0.05}$ | $\mathbf{2.73}_{\pm 0.04}$ | $\mathbf{2.74}_{\pm 0.03}$ |
| energy | $0.46_{\pm 0.04}$ | $0.47_{\pm 0.04}$ | $0.64_{\pm 0.02}$ | $0.79_{\pm 0.03}$ | $\mathbf{0.31}_{\pm 0.03}$ | $0.67_{\pm 0.01}$ | $0.78_{\pm 0.01}$ |
| kin8nm | $-1.34_{\pm 0.01}$ | $-1.32_{\pm 0.01}$ | $-1.30_{\pm 0.00}$ | $-1.29_{\pm 0.01}$ | $\mathbf{-1.38}_{\pm 0.01}$ | $-1.36_{\pm 0.01}$ | $-1.33_{\pm 0.01}$ |
| naval | $-6.61_{\pm 0.03}$ | $-6.65_{\pm 0.05}$ | $-5.93_{\pm 0.04}$ | $-5.83_{\pm 0.03}$ | $-6.51_{\pm 0.03}$ | $-5.95_{\pm 0.03}$ | $-5.80_{\pm 0.01}$ |
| power | $2.66_{\pm 0.01}$ | $2.66_{\pm 0.01}$ | $2.62_{\pm 0.01}$ | $2.62_{\pm 0.01}$ | $\mathbf{2.57}_{\pm 0.01}$ | $\mathbf{2.56}_{\pm 0.00}$ | $2.60_{\pm 0.00}$ |
| protein | $2.29_{\pm 0.00}$ | $2.22_{\pm 0.01}$ | $2.22_{\pm 0.01}$ | $2.31_{\pm 0.01}$ | $2.05_{\pm 0.01}$ | $2.18_{\pm 0.01}$ | $2.29_{\pm 0.01}$ |
| wine | $0.89_{\pm 0.04}$ | $0.90_{\pm 0.03}$ | $-0.36_{\pm 0.03}$ | $0.15_{\pm 0.02}$ | $0.80_{\pm 0.03}$ | $-0.19_{\pm 0.02}$ | $0.11_{\pm 0.02}$ |
| yacht | $0.07_{\pm 0.08}$ | $0.07_{\pm 0.06}$ | $0.62_{\pm 0.04}$ | $0.93_{\pm 0.04}$ | $\mathbf{-0.16}_{\pm 0.05}$ | $0.80_{\pm 0.03}$ | $0.92_{\pm 0.03}$ |

Table 7: RMSE results on the UCI Datasets benchmark (lower is better). Scores that perform at least on par when compared to the results in Table 2 are highlighted in **bold**.

| dataset | CRPS | WCRPS | WCRPS$^{mh}$ | WCRPS$^{lmh}$ | WCRPS$_e$ | WCRPS$_e^{mh}$ | WCRPS$_e^{lmh}$ |
|---|---|---|---|---|---|---|---|
| boston | $3.01_{\pm 0.21}$ | $3.07_{\pm 0.23}$ | $3.08_{\pm 0.20}$ | $3.12_{\pm 0.25}$ | $\mathbf{2.91}_{\pm 0.18}$ | $2.98_{\pm 0.20}$ | $2.93_{\pm 0.20}$ |
| concrete | $5.48_{\pm 0.16}$ | $5.51_{\pm 0.14}$ | $5.11_{\pm 0.15}$ | $5.23_{\pm 0.15}$ | $\mathbf{4.94}_{\pm 0.17}$ | $\mathbf{4.87}_{\pm 0.15}$ | $\mathbf{4.85}_{\pm 0.13}$ |
| energy | $0.44_{\pm 0.02}$ | $0.45_{\pm 0.02}$ | $0.52_{\pm 0.02}$ | $0.48_{\pm 0.02}$ | $\mathbf{0.41}_{\pm 0.01}$ | $0.45_{\pm 0.02}$ | $0.43_{\pm 0.01}$ |
| kin8nm | $0.07_{\pm 0.00}$ | $0.07_{\pm 0.00}$ | $0.07_{\pm 0.00}$ | $0.07_{\pm 0.00}$ | $\mathbf{0.06}_{\pm 0.00}$ | $\mathbf{0.06}_{\pm 0.00}$ | $0.07_{\pm 0.00}$ |
| naval | $0.00_{\pm 0.00}$ | $0.00_{\pm 0.00}$ | $0.00_{\pm 0.00}$ | $0.00_{\pm 0.00}$ | $0.00_{\pm 0.00}$ | $0.00_{\pm 0.00}$ | $0.00_{\pm 0.00}$ |
| power | $3.79_{\pm 0.05}$ | $3.79_{\pm 0.04}$ | $3.74_{\pm 0.04}$ | $3.71_{\pm 0.04}$ | $\mathbf{3.62}_{\pm 0.04}$ | $3.65_{\pm 0.04}$ | $3.64_{\pm 0.04}$ |
| protein | $3.77_{\pm 0.01}$ | $3.76_{\pm 0.02}$ | $3.86_{\pm 0.02}$ | $3.80_{\pm 0.03}$ | $\mathbf{3.47}_{\pm 0.02}$ | $3.69_{\pm 0.01}$ | $3.60_{\pm 0.04}$ |
| wine | $0.64_{\pm 0.01}$ | $0.64_{\pm 0.01}$ | $0.64_{\pm 0.01}$ | $0.65_{\pm 0.01}$ | $\mathbf{0.63}_{\pm 0.01}$ | $\mathbf{0.63}_{\pm 0.01}$ | $0.64_{\pm 0.01}$ |
| yacht | $0.95_{\pm 0.11}$ | $0.96_{\pm 0.13}$ | $0.89_{\pm 0.09}$ | $1.05_{\pm 0.12}$ | $\mathbf{0.78}_{\pm 0.09}$ | $0.99_{\pm 0.10}$ | $0.94_{\pm 0.11}$ |

1. They do not use the same train-test splits as other approaches mentioned in Section 5.3, rather random ones;

2. Their code appears to validate on the test set, and reports *the minimum of the achieved NLL results* over all epochs.

The setup seems consistent across all measured methods in the paper, therefore their results are internally consistent (although an argument could be made that it favors methods with good predictions *and large variance* across training epochs). However, unfortunately their results are not comparable with the rest of the literature as presented.

Implementing a proper train-validation-test setup for their model appears to result in subpar performance in our experiments, however this could be the case for a number of reasons, such as suboptimal hyperparameter choices on our part.

## C.7 CRPS MODEL VARIANTS ON THE UCI BENCHMARK

We present NLL scores in Table 6 and RMSE scores in Table 7 for various CRPS-based model variants. A subscript of *e* indicates ensemble models, superscript of *mh* and *lmh* indicates multi-head and layered-multi-head variants, respectively.

## C.8 RUNNING TIMES & SCALING

See Table 8 for running times in seconds on the UCI Benchmark. All of our implementations were run on an AMD EPYC 7F72 workstation with 2x24 CPU cores (48 cores, 96 threads in total) and 384 MiB of L3 cache. We used a single NVIDIA A100-SXM4-40GB GPU per experiment, with CUDA version 12.2 and driver version 535.183.01, ensuring consistent conditions across experiments.

To illustrate scaling, we compare the running time of training 10 epochs with MDN and CRPS on the largest *protein* dataset for increasingly large model sizes, however leaving the expressive power

Table 8: Running times on the full UCI Benchmark in seconds.

| MDN | WCRPS | Ensembles | $WCRPS_{ens}$ | $MDN_{bnn}$ |
|---|---|---|---|---|
| 5594 | 12074 | 19070 | 39624 | 95902 |

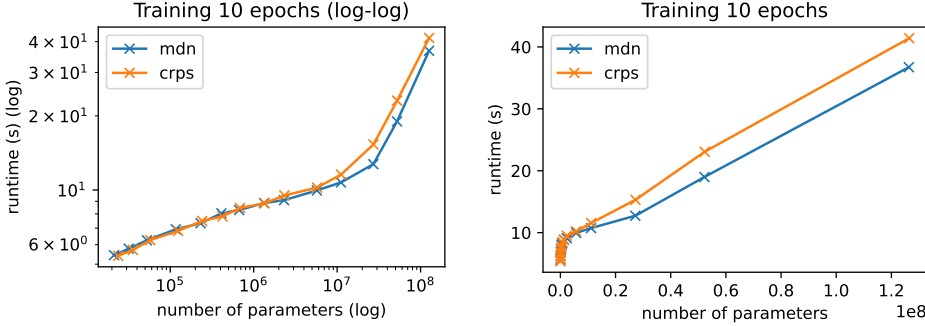

Figure 7: Scaling of a CRPS badsed models compared to a deterministic model.

w.r.t. the output distribution constant (100 components for MDN and two layers of size [100, 50] for the CRPS based model). See Figure 7 for results.

At smaller model sizes, the specifics of the GPU hardware make comparisons somewhat random, as devices can execute a large amount of computation in one step. However, as expected, at larger sizes, the nondeterministic part of the network incurs a constant amount of additional cost, leaving the running time approximately linear in the number of model parameters.

## C.9    CALIBRATION

Please refer to Figure 8 for calibration plots on all train and test datasets.

## C.10    DATASET STATISTICS

Please refer to Table 9 for dataset statistics.

## C.11    PERFORMANCE AGAINST SIMPLE METHODS

While the primary concern in term of performance is comparison against advanced alternatives, such as the baselines of Section 5.3, there is value in observing the improvement when comparing against more traditional model types. In this section we present measurements comparing our

Table 9: Value frequencies in the UCI benchmark datasets

| dataset | samples ($m$) | unique y | unique / $m$ | most frequent y | most frequent / $m$ |
|---|---|---|---|---|---|
| boston | 506 | 229 | 45.26% | 16 | 3.16% |
| concrete | 1030 | 845 | 82.04% | 6 | 0.58% |
| energy | 768 | 586 | 76.30% | 6 | 0.78% |
| kin8nm | 8192 | 8191 | 99.99% | 2 | 0.02% |
| naval | 11934 | 51 | 0.43% | 234 | 1.96% |
| power | 9568 | 4836 | 50.54% | 9 | 0.09% |
| protein | 45730 | 15903 | 34.78% | 272 | 0.59% |
| wine | 1599 | 6 | 0.38% | 681 | 42.59% |
| yacht | 308 | 258 | 83.77% | 3 | 0.97% |

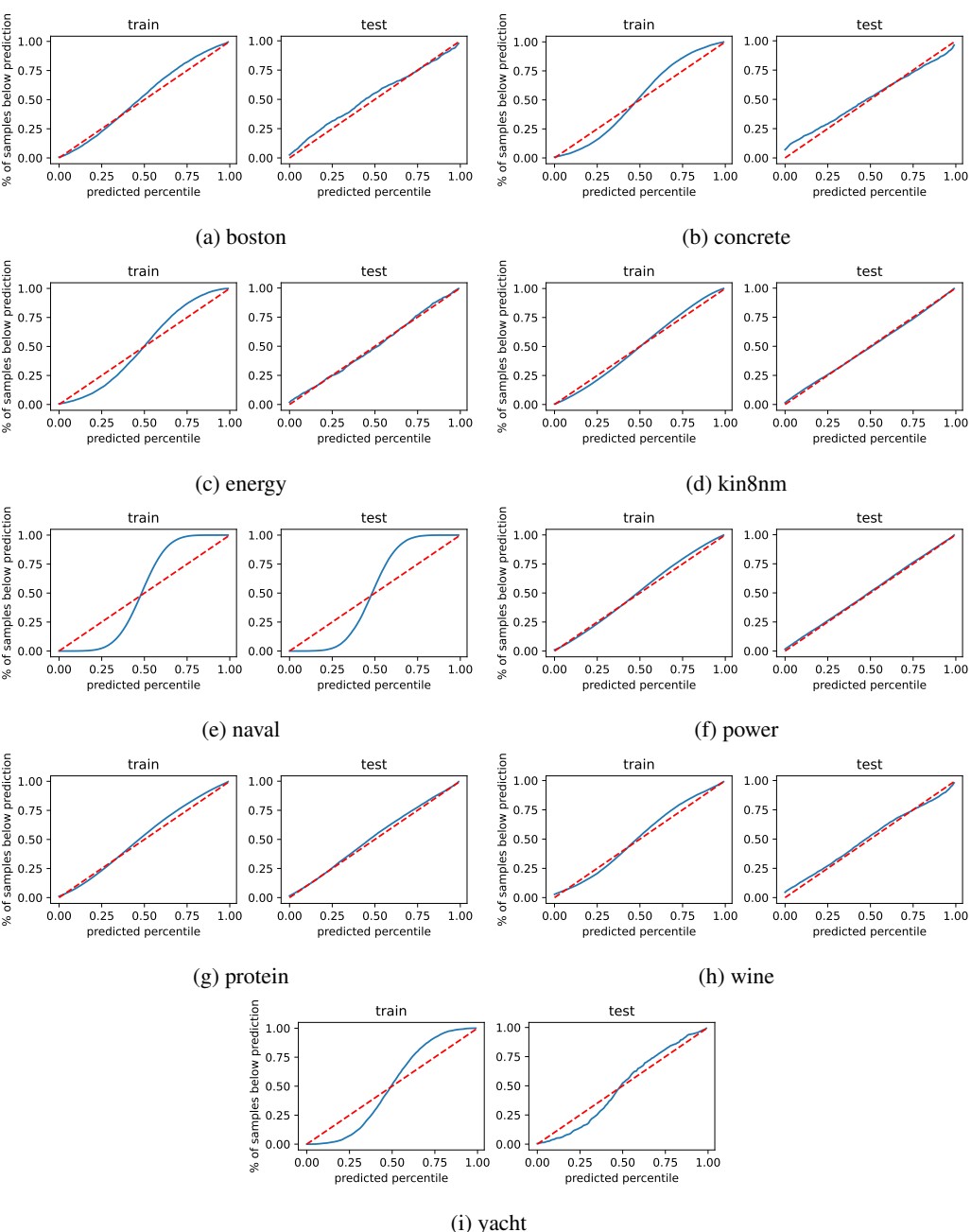

Figure 8: Calibration for the WCRPS$_e$ model on all datasets separately for train and test sets

Table 10: NLL results on the UCI Datasets benchmark (lower is better). Scores that perform at least on par when compared to the results in Table 1 are highlighted in **bold**.

| dataset | SHR | MDN | BNN | MDN$_{bnn}$ | WCRPS$_e$ |
|---|---|---|---|---|---|
| boston | $2.74_{\pm0.11}$ | $2.45_{\pm0.06}$ | $2.36_{\pm0.04}$ | $2.36_{\pm0.03}$ | $\mathbf{2.32}_{\pm0.05}$ |
| concrete | $3.30_{\pm0.09}$ | $3.16_{\pm0.03}$ | $2.98_{\pm0.03}$ | $2.99_{\pm0.03}$ | $2.95_{\pm0.05}$ |
| energy | $1.53_{\pm0.21}$ | $1.09_{\pm0.06}$ | $1.00_{\pm0.21}$ | $0.65_{\pm0.02}$ | $\mathbf{0.31}_{\pm0.03}$ |
| kin8nm | $-1.20_{\pm0.01}$ | $-1.15_{\pm0.01}$ | $-1.28_{\pm0.01}$ | $-1.30_{\pm0.01}$ | $\mathbf{-1.38}_{\pm0.01}$ |
| naval | $-6.59_{\pm0.03}$ | $-6.55_{\pm0.03}$ | $\mathbf{-6.84}_{\pm0.03}$ | $\mathbf{-6.72}_{\pm0.01}$ | $-6.51_{\pm0.03}$ |
| power | $2.78_{\pm0.02}$ | $2.67_{\pm0.00}$ | $2.77_{\pm0.01}$ | $2.62_{\pm0.01}$ | $\mathbf{2.57}_{\pm0.01}$ |
| protein | $2.89_{\pm0.06}$ | $2.02_{\pm0.01}$ | $2.73_{\pm0.02}$ | $\mathbf{1.96}_{\pm0.01}$ | $2.05_{\pm0.01}$ |
| wine | $0.98_{\pm0.03}$ | $-3.75_{\pm0.06}$ | $0.89_{\pm0.02}$ | $\mathbf{-5.11}_{\pm0.09}$ | $0.80_{\pm0.03}$ |
| yacht | $0.67_{\pm0.08}$ | $0.79_{\pm0.09}$ | $0.26_{\pm0.07}$ | $0.63_{\pm0.04}$ | $\mathbf{-0.16}_{\pm0.05}$ |

Table 11: RMSE results on the UCI Datasets benchmark (lower is better). Scores that perform at least on par when compared to the results in Table 2 are highlighted in **bold**.

| dataset | SHR | MDN | BNN | MDN$_{bnn}$ | WCRPS$_e$ |
|---|---|---|---|---|---|
| boston | $3.63_{\pm0.25}$ | $3.54_{\pm0.27}$ | $3.01_{\pm0.24}$ | $2.93_{\pm0.20}$ | $\mathbf{2.91}_{\pm0.18}$ |
| concrete | $6.20_{\pm0.16}$ | $6.34_{\pm0.11}$ | $5.30_{\pm0.12}$ | $5.25_{\pm0.11}$ | $\mathbf{4.94}_{\pm0.17}$ |
| energy | $1.51_{\pm0.20}$ | $1.89_{\pm0.13}$ | $0.54_{\pm0.03}$ | $0.51_{\pm0.01}$ | $\mathbf{0.41}_{\pm0.01}$ |
| kin8nm | $0.08_{\pm0.00}$ | $0.08_{\pm0.00}$ | $0.07_{\pm0.00}$ | $0.07_{\pm0.00}$ | $\mathbf{0.06}_{\pm0.00}$ |
| naval | $0.00_{\pm0.00}$ | $0.00_{\pm0.00}$ | $0.00_{\pm0.00}$ | $0.00_{\pm0.00}$ | $0.00_{\pm0.00}$ |
| power | $3.94_{\pm0.04}$ | $3.96_{\pm0.04}$ | $3.96_{\pm0.04}$ | $3.86_{\pm0.04}$ | $\mathbf{3.62}_{\pm0.04}$ |
| protein | $4.54_{\pm0.04}$ | $4.07_{\pm0.03}$ | $4.39_{\pm0.04}$ | $4.09_{\pm0.02}$ | $\mathbf{3.47}_{\pm0.02}$ |
| wine | $0.64_{\pm0.01}$ | $0.65_{\pm0.01}$ | $\mathbf{0.63}_{\pm0.01}$ | $0.66_{\pm0.01}$ | $\mathbf{0.63}_{\pm0.01}$ |
| yacht | $1.39_{\pm0.17}$ | $1.68_{\pm0.18}$ | $0.88_{\pm0.08}$ | $\mathbf{0.77}_{\pm0.07}$ | $0.78_{\pm0.09}$ |

method against a simple probabilistic neural-network based heteroscedastic regression model, and a classic BNN trained with variational inference.

A simple heteroscedastic regression model predicts the mean along with the variance of the model, and computes the loss function accordingly (**SHR** for **S**imple **H**eteroscedastic **R**egression). Note that assuming normal distribution for the noise, this model coincides with MDN when setting the number of components to 1.

Classic Bayesian neural networks also predict the parameters of the aleatoric distribution, however also assume probability distributions over the weights of the network. The resulting network can be optimized using variational inference Blundell et al. (2015) (**BNN**). Again assuming normal distribution for the noise, this model coincides with a BNN-based version of MDN with 1 components.

We present results for SHR, MDN, BNN, MDN$_{bnn}$, and WCRPS$_e$ in Tables 10 and 11. As we can observe, in most cases simple methods either benefit from relaxing the Gaussian assumption, or perform similarly to the corresponding variant of MDN. On some datasets, however, the Gaussian restriction *does* seem to benefit the method, most apparent on the *yacht* and *naval* datasets.

### C.12   PERFORMANCE COMPARISON FOR THE NUMBER OF SAMPLES

As the CRPS-based methods represent their distribution through producing samples, the number of samples drawn for calculating the loss is an important hyperparameter during training. In Table 12, we present performance as well as running times for various choices for the number of samples drawn during training.

A smaller number of samples drawn results in faster per-batch calculation time, but in turn the training process takes longer to converge. While the running time overall seems to vary randomly to a relatively large degree, still we can observe that smaller values result in faster running times.

Table 12: NLL results on the UCI Datasets benchmark (lower is better) for different values of samples drawn during training.

| dataset | 2 | 3 | 5 | 10 | 20 | 50 | 100 | 200 |
|---|---|---|---|---|---|---|---|---|
| boston | $2.33_{\pm0.04}$ | $2.30_{\pm0.04}$ | $2.40_{\pm0.04}$ | $2.39_{\pm0.05}$ | $2.35_{\pm0.05}$ | $2.33_{\pm0.06}$ | $2.40_{\pm0.05}$ | $2.40_{\pm0.05}$ |
| concrete | $3.02_{\pm0.03}$ | $3.04_{\pm0.04}$ | $3.06_{\pm0.05}$ | $3.04_{\pm0.04}$ | $3.08_{\pm0.03}$ | $3.09_{\pm0.04}$ | $3.09_{\pm0.04}$ | $3.11_{\pm0.05}$ |
| energy | $0.59_{\pm0.03}$ | $0.49_{\pm0.02}$ | $0.49_{\pm0.03}$ | $0.47_{\pm0.03}$ | $0.50_{\pm0.05}$ | $0.44_{\pm0.03}$ | $0.47_{\pm0.04}$ | $0.46_{\pm0.05}$ |
| kin8nm | $-1.24_{\pm0.02}$ | $-1.34_{\pm0.01}$ | $-1.34_{\pm0.01}$ | $-1.34_{\pm0.01}$ | $-1.34_{\pm0.01}$ | $-1.33_{\pm0.01}$ | $-1.32_{\pm0.01}$ | $-1.33_{\pm0.01}$ |
| naval | $-6.07_{\pm0.03}$ | $-6.23_{\pm0.03}$ | $-6.39_{\pm0.03}$ | $-6.41_{\pm0.03}$ | $-6.52_{\pm0.04}$ | $-6.57_{\pm0.03}$ | $-6.65_{\pm0.05}$ | $-6.58_{\pm0.03}$ |
| power | $2.81_{\pm0.01}$ | $2.80_{\pm0.01}$ | $2.78_{\pm0.00}$ | $2.73_{\pm0.01}$ | $2.66_{\pm0.01}$ | $2.66_{\pm0.01}$ | $2.66_{\pm0.01}$ | $2.65_{\pm0.01}$ |
| protein | $2.66_{\pm0.01}$ | $2.60_{\pm0.01}$ | $2.51_{\pm0.01}$ | $2.27_{\pm0.04}$ | $2.21_{\pm0.01}$ | $2.22_{\pm0.00}$ | $2.22_{\pm0.01}$ | $2.23_{\pm0.01}$ |
| wine | $0.93_{\pm0.02}$ | $0.91_{\pm0.03}$ | $0.92_{\pm0.02}$ | $0.91_{\pm0.03}$ | $0.92_{\pm0.03}$ | $0.92_{\pm0.02}$ | $0.90_{\pm0.03}$ | $0.87_{\pm0.03}$ |
| yacht | $0.23_{\pm0.05}$ | $0.13_{\pm0.05}$ | $0.05_{\pm0.06}$ | $-0.05_{\pm0.06}$ | $0.00_{\pm0.05}$ | $-0.08_{\pm0.07}$ | $0.07_{\pm0.06}$ | $0.08_{\pm0.08}$ |
| time (h) | 2.34 | 2.51 | 2.41 | 2.90 | 3.61 | 3.55 | 3.35 | 3.43 |

On the other hand, while some smaller datasets seem to favor small sample values (notably *boston* and *concrete*), larger sample sizes seem much more robust overall, especially on the larger datasets in the benchmark (*naval, power, protein*).

## C.13 PERFORMANCE COMPARISON AGAINST DISCO NETS

As described before, the closest prior approach is DISCO Nets (Bouchacourt et al., 2016), where the authors introduce a multivariate distributional regression method for hand-pose estimation. Remarkably, they also use injected random noise for nondeterminism (as in Section 4.1), and derive a loss function which, with the correct parametrization, is very similar to a multivariate version of the basic non-weighted version of our method, essentially reproducing the Energy Score (see Section 4.5), which the authors also highlight.

As the DISCO Nets method is not primarily formulated for univariate regression, for fairness we first compare on the original task of (Bouchacourt et al., 2016). DISCO Nets is originally measured on the NYU Hand Pose Estimation dataset (Tompson et al., 2014) as preprocessed by Oberweger et al. (2015). The processed dataset contains 72,757 training and 8254 testing frames, with the sets consisting of RGBD images of hands, taken from 3 viewpoints. However, only the front view and the depth channel is used for the experiment. For each data point, the label consist of the 3d position of 14 finger joints.

To compare, we use the original code of DISCO Nets[6] to run the original preprocessing pipeline to acquire the train and test sets. We also use the same network size as Bouchacourt et al. (2016), consisting of 3 convolutional layers with 8 filters each, max pooling layers, and 3 linear layers of size 1024 with ReLU activation.

Performance is measured as in Bouchacourt et al. (2016), in terms of Energy Score (denoted as *ProbLoss*, smaller is better), Mean Joint Euclidean Error (MeJEE, smaller is better), Max Joint Euclidean Error (MaJEE, smaller is better) and Fraction of Frames within distance (FF, larger is better), including standard error of the mean (SEM) for the first three. For exact definitions, please refer to the original paper and its supplement. We take scores for DISCO Nets and baselines from the original paper, and present scores for WCRPS and WCRSP$_e$ in Table 13. As we can observe, the improved models outperform DISCO Nets, with the WCRPS$_e$ model performing really strongly in particular. While deterministic measures (MeJEE, MaJEE, FF) also show higher scores, the performance difference is most apparent in ProbLoss, hinting at much better predictive distributions.

Second, we implement the method on the UCI Benchmark and evaluate its performance under the same conditions as in Section 5.3.1. For hyperparameters not explicitly investigated in Bouchacourt et al. (2016), we adopt the same values as the original authors (e.g., $\beta = 1$). Since DISCO Nets also generates samples rather than directly modeling the density, we apply the KDE approach described in Appendix C.1 to compute the NLL.

---

[6]https://github.com/oval-group/DISCONets

Table 13: Performance comparison against Bouchacourt et al. (2016) and its baselines using the multivariate WCRPS and WCRPS$_w$ models on the NYU Hand Pose dataset, $\pm$SEM.

| Model | ProbLoss (mm) | MeJEE (mm) | MaJEE (mm) | FF (80mm) |
|---|---|---|---|---|
| BASE$_{\beta=1,\sigma=1}$ | 103.8$\pm$0.6 | 25.2$\pm$0.2 | 52.7$\pm$0.2 | 86.040 |
| BASE$_{\beta=1,\sigma=5}$ | 99.3$\pm$0.6 | 25.5$\pm$0.2 | 52.9$\pm$0.3 | 85.773 |
| BASE$_{\beta=1,\sigma=10}$ | 96.3$\pm$0.6 | 25.7$\pm$0.1 | 53.2$\pm$0.3 | 85.664 |
| DISCO$_{\beta=1,\gamma=0}$ | 92.9$\pm$0.5 | 21.6$\pm$0.1 | 46.0$\pm$0.3 | 92.971 |
| DISCO$_{\beta=1,\gamma=0.25}$ | 89.9$\pm$0.5 | 21.2$\pm$0.1 | 46.4$\pm$0.3 | 93.262 |
| DISCO$_{\beta=1,\gamma=0.5}$ | 83.8$\pm$0.5 | 20.9$\pm$0.1 | 45.1$\pm$0.2 | 94.438 |
| WCRPS | 72.1$\pm$0.5 | 19.5$\pm$0.1 | 39.0$\pm$0.2 | 96.510 |
| WCRPS$_e$ | **63.7$\pm$0.5** | **18.8$\pm$0.1** | **37.6$\pm$0.2** | **96.958** |

Table 14: NLL results on the UCI Datasets benchmark (lower is better). Scores that perform at least on par when compared to the results in Table 1 are highlighted in **bold**.

| dataset | dropout | CRPS | WCRPS$_e$ | DISCO$^{\beta=1}_{\gamma=0}$ | DISCO$^{\beta=1}_{\gamma=0.25}$ | DISCO$^{\beta=1}_{\gamma=0.5}$ | BNN-LV |
|---|---|---|---|---|---|---|---|
| boston | 2.40$_{\pm0.04}$ | 2.38$_{\pm0.06}$ | **2.32**$_{\pm0.05}$ | 3.44$_{\pm0.14}$ | 2.86$_{\pm0.09}$ | 2.34$_{\pm0.04}$ | 2.68$_{\pm0.02}$ |
| concrete | **2.94**$_{\pm0.02}$ | 3.11$_{\pm0.04}$ | 2.95$_{\pm0.05}$ | 4.16$_{\pm0.06}$ | 3.51$_{\pm0.04}$ | 3.02$_{\pm0.03}$ | 3.31$_{\pm0.01}$ |
| energy | 1.21$_{\pm0.01}$ | 0.46$_{\pm0.04}$ | **0.31**$_{\pm0.03}$ | 2.61$_{\pm0.11}$ | 0.82$_{\pm0.12}$ | 0.65$_{\pm0.03}$ | 2.32$_{\pm0.02}$ |
| kin8nm | -1.14$_{\pm0.01}$ | -1.34$_{\pm0.01}$ | **-1.38**$_{\pm0.01}$ | 0.81$_{\pm0.03}$ | -0.83$_{\pm0.01}$ | -1.33$_{\pm0.01}$ | -1.24$_{\pm0.01}$ |
| naval | -4.45$_{\pm0.00}$ | -6.61$_{\pm0.03}$ | -6.51$_{\pm0.03}$ | -6.59$_{\pm0.03}$ | -6.47$_{\pm0.03}$ | -6.04$_{\pm0.03}$ | -5.14$_{\pm0.01}$ |
| power | 2.81$_{\pm0.01}$ | 2.66$_{\pm0.01}$ | **2.57**$_{\pm0.01}$ | 3.54$_{\pm0.03}$ | 3.13$_{\pm0.01}$ | 2.71$_{\pm0.01}$ | 2.83$_{\pm0.01}$ |
| protein | 2.87$_{\pm0.00}$ | 2.29$_{\pm0.00}$ | 2.05$_{\pm0.01}$ | 5.30$_{\pm0.14}$ | inf$_{\pm\text{nan}}$ | 2.34$_{\pm0.01}$ | 2.59$_{\pm0.01}$ |
| wine | 0.93$_{\pm0.01}$ | 0.89$_{\pm0.04}$ | 0.80$_{\pm0.03}$ | 2.62$_{\pm0.09}$ | 1.75$_{\pm0.06}$ | 0.94$_{\pm0.03}$ | 1.00$_{\pm0.01}$ |
| yacht | 1.25$_{\pm0.02}$ | 0.07$_{\pm0.08}$ | **-0.16**$_{\pm0.05}$ | 0.29$_{\pm0.07}$ | 0.10$_{\pm0.07}$ | 0.37$_{\pm0.07}$ | 2.09$_{\pm0.03}$ |

In Bouchacourt et al. (2016), the authors propose using the MEU (Maximum Expected Utility) point, conceptually similar to the geometric median, and suggest employing MEU-based RMSE for early stopping by comparing it to the ground truth. However, in our experiments, we resort to using NLL for early stopping, as MEU-based RMSE proved ineffective; it often failed to decrease beyond a certain threshold or even increased from its starting point. In a univariate regression problem, the output distribution's geometric median is equivalent to the median, thus MEU is essentially defined as the median of the sample. Given that the median generally does not coincide with the mean which is known to be optimal for RMSE, this outcome is not unexpected.

We test three hyperparameter configurations for DISCO Nets, as presented in Tables 14 and 15. As expected, the theoretically optimal variant ($\gamma = 0.5$) performs well, while other configurations yield suboptimal results. However, even the correctly parameterized version is outperformed by our best method. This outcome is again unsurprising, since with the correct hyperparameter settings, adaptations for regression benchmarking, and other enhancements we apply for training, the method essentially replicates the unweighted CRPS approach trained using a small sample size for loss calculation.

## C.14 PERFORMANCE COMPARISON AGAINST BNN-LV

The method introduced in Depeweg et al. (2018) (BNN-LV) is theoretically capable of representing complex distributions, such multi-modal ones. However, the original paper primarily evaluates the method on reinforcement learning tasks and lacks detailed quantitative analysis. Therefore, we choose not to benchmark against it using its original datasets. Fortunately, a recent uncertainty quantification framework (Lehmann et al., 2024) implements BNN-LV for regression. Given that the first author of Depeweg et al. (2018) is also a co-author of Lehmann et al. (2024) and was

---

[6]When a ground truth point falls too far outside the predicted distribution, the likelihood function becomes essentially 0 numerically, causing the NLL metric to take on the value of $\infty$.

Table 15: RMSE results on the UCI Datasets benchmark (lower is better). Scores that perform at least on par when compared to the results in Table 2 are highlighted in **bold**.

| dataset | dropout | CRPS | WCRPS$_e$ | DISCO$_{\gamma=0}^{\beta=1}$ | DISCO$_{\gamma=0.25}^{\beta=1}$ | DISCO$_{\gamma=0.5}^{\beta=1}$ | BNN-LV |
|---------|---------|------|-----------|------------------------------|----------------------------------|---------------------------------|--------|
| boston | $3.61_{\pm0.23}$ | $3.01_{\pm0.21}$ | $\mathbf{2.91}_{\pm0.18}$ | $4.15_{\pm0.27}$ | $3.49_{\pm0.21}$ | $2.99_{\pm0.20}$ | $3.67_{\pm0.23}$ |
| concrete | $5.45_{\pm0.19}$ | $5.48_{\pm0.16}$ | $\mathbf{4.94}_{\pm0.17}$ | $6.95_{\pm0.12}$ | $6.34_{\pm0.17}$ | $5.36_{\pm0.17}$ | $6.54_{\pm0.11}$ |
| energy | $0.97_{\pm0.06}$ | $0.44_{\pm0.02}$ | $\mathbf{0.41}_{\pm0.01}$ | $2.87_{\pm0.14}$ | $0.71_{\pm0.15}$ | $0.56_{\pm0.02}$ | $3.00_{\pm0.08}$ |
| kin8nm | $0.09_{\pm0.00}$ | $0.07_{\pm0.00}$ | $\mathbf{0.06}_{\pm0.00}$ | $0.08_{\pm0.00}$ | $0.07_{\pm0.00}$ | $0.07_{\pm0.00}$ | $0.07_{\pm0.00}$ |
| naval | $0.00_{\pm0.00}$ | $0.00_{\pm0.00}$ | $0.00_{\pm0.00}$ | $0.00_{\pm0.00}$ | $0.00_{\pm0.00}$ | $0.00_{\pm0.00}$ | $0.00_{\pm0.00}$ |
| power | $4.18_{\pm0.04}$ | $3.79_{\pm0.05}$ | $\mathbf{3.62}_{\pm0.04}$ | $5.23_{\pm0.12}$ | $4.55_{\pm0.04}$ | $3.95_{\pm0.04}$ | $4.17_{\pm0.03}$ |
| protein | $4.39_{\pm0.02}$ | $3.77_{\pm0.01}$ | $\mathbf{3.47}_{\pm0.02}$ | $5.75_{\pm0.01}$ | $4.70_{\pm0.08}$ | $3.96_{\pm0.03}$ | $4.79_{\pm0.02}$ |
| wine | $0.66_{\pm0.01}$ | $0.64_{\pm0.01}$ | $\mathbf{0.63}_{\pm0.01}$ | $0.70_{\pm0.01}$ | $0.65_{\pm0.01}$ | $0.65_{\pm0.01}$ | $0.64_{\pm0.01}$ |
| yacht | $1.23_{\pm0.37}$ | $0.95_{\pm0.11}$ | $0.78_{\pm0.09}$ | $1.07_{\pm0.11}$ | $0.88_{\pm0.10}$ | $1.05_{\pm0.10}$ | $3.38_{\pm0.20}$ |

substantially involved in the BNN-LV implementation[7], we consider the framework of Lehmann et al. (2024) to be an authoritative reference implementation for BNN-LV.

For NLL calculations, we use the KDE approach outlined in Appendix C.1, and early stopping is performed based on measuring the training loss on the validation set. As the method converges very slowly, we needed to increase the maximum number of allowed epochs. However, since it also requires relatively long per-epoch training times, the approach ended up taking approximately 55X as long as our fastest measured method MDN (or over 86 hours) on the full benchmark, making it impractical to optimize over multiple hyperparameter combinations. Instead, we employ the hyperparameters suggested in the framework's manual. The results are summarized in Tables 14 and 15.

On many datasets, BNN-LV produces respectable scores, often comparable to classic baselines such as *dropout* in NLL, however often underperforms in RMSE. In general it does not produce competitive scores when compared to the best measured non-Gaussian methods such as WCRPS$_e$ or MDN$_{\text{bnn}}$.

---

[7]https://github.com/lightning-uq-box/lightning-uq-box/pull/21

