# OpenReview forum: "Distribution-Free Data Uncertainty for Neural Network Regression"
_ICLR.cc/2025/Conference — ICLR 2025 Poster_

### Official Review · Reviewer_cN1m · 2024-11-02

**Soundness:** 4
**Presentation:** 4
**Contribution:** 3
**Rating:** 8
**Confidence:** 5

**Summary:**

The authors approach the task of modeling irreducible (aleatoric) uncertainty in deep regression models. To do this, they rely on the _continuous ranked probability score (CPRS)_ as the target loss rather than, e.g., a parametric likelihood. They introduce an efficient unbiased estimator to the CRPS with a cost of $O(n\log(n))$ compared to a naive $O(n^2)$ approach.
The method is evaluated on a range of regression tasks.


_____
_Update: Given the changes to the submission provided during the rebuttal discussion, I am increasing my score accordingly._

**Strengths:**

- The paper is well-written and easy to follow.
- The method is well-motivated and the proposed CRPS-based approach provides a clear and straightforward approach to settings where it is beneficial to move away from more common unimodal likelihood assumptions (although see below).
- The proposed approach is computationally scalable.
- The appendix and provided codebase seem to ensure full replicability. (However, I didn't check the code in detail and only skimmed it.)
- At least on the rather simple UCI regression tasks the model is well calibrated. It would have been interesting to see regression problems that require deeper nets, e.g., an evaluation of a depth-estimation data set from the computer vision literature.

**Weaknesses:**

- The major weakness is in the framing of the storyline. Again and again throughout the text, the authors claim that aleatoric has not been of interest to the literature or has been only _"treated as an afterthought"_ (l180).
I strongly disagree with this statement. Well-modeled aleatoric uncertainty has a strong presence in the literature, as is, e.g., discussed in the cited review paper by Gawlikowski et al. (2023).
Be it in computer vision (Kendall & Gal, 2017; Gast & Roth, 2018,...), any normal likelihood-based paper that learns a homoscedastic or heteroscedastic noise model, evidential deep learning/prior network-based approaches such as Amini et al. (2020) or Malinin et al. (2020), and many more.
Accurately modeling both aleatoric and epistemic uncertainty has, e.g., the important application of active learning, where information should be requested about data the model is uncertain in a reducible rather than an irreducible sense and other approaches.
As such, aleatoric uncertainty is not an afterthought, but rather an integral component of those models. However, where I agree, and which would be a more correct phrasing of the problem, is that most of these focus on an assumption of a univariate noise structure rather than a multi-modal one as the authors allow for.
Yet, this constraint is not an inherent one or one of ignorance in the literature. Lakshminarayanan et al. (2017), e.g., explicitly mention the proposal of extending their method with MDNs if a uni-modal normal likelihood is too restrictive.
This restriction is also regularly lifted in the literature. See, e.g., the work of Depeweg et al. (2018), who allow for multi-modality via latent variables to the input of BNNs, or Bouchacourt et al.'s (2016) DISCO nets, who inject noise in a later part of the deterministic nets, in an architectural similarity with the proposal by the authors.
Non-normal likelihoods have also been the focus of, e.g., Gaussian Processes (Sendera et al., 2021), Normalizing Flows (e.g., Ardizzone et al. 2019), and other probabilistic models.
The authors' proposal is innovative and novel, but not unique in the sense that nobody has cared about aleatoric uncertainty since Bishop (1994).
This last sentence is a slightly exaggerated version of how parts of the paper read. I don't claim that the authors think that, but both the discussion and experimental evaluation ignore this field almost completely.
- The discussion on how the method fits into the literature, and that the proposal is primarily on nonparametric vs unimodal noise structure should be clearer.
- Given this, an experimental comparison against, e.g., a naive heteroscedastic (deterministic or Bayesian) neural net, DISCO Nets, or Depeweg et al. (2018) is necessary to evaluate the method properly.


_____
Adrizzone et al., Analyzing inverse problems with invertible neural networks, ICLR 2019
Bouchacourt et al., DISCO Nets: DISSimilarity COefficient Networks, NeurIPS 2016
Depeweg et al., Decomposition of Uncertainty in Bayesian Deep Learning for Efficient and Risk-sensitive Learning, ICML 2018
Gast & Roth, Lightweight Probabilistic Deep Networks, CVPR 2018
Malinin et al. Regression Prior Networks, 2020
Sendera et al., Non-Gaussian Processes for Few-Shot Regression, NeurIPS 2021

**Questions:**

- Q1: Can the authors comment to a greater degree on the relationship between their proposal and the existing literature?
- Q2: All networks in the paper are rather small. How do the authors expect their method to scale as the architectural structure becomes deeper?

---

> ### Author Response · Authors · 2024-11-18
>
> Thank you for your review! We really appreciate the time and thought you put into your comments, which we find very insightful and useful. We were also honored to read your kind words regarding the paper in general. We respond to the listed weaknesses and questions in separate points below.
>
> __Weaknesses__
> - We see your point and really appreciate you raising this issue. In general, our intention was to highlight the fact that *specifically for neural network regression*, the *representation* of aleatoric uncertainty is an under-researched area. Thus our intent is indeed more aligned with the second interpretation in your comment (unimodal vs multimodal noise structures). Though we prefer to be a bit more general in this area, and also include other types of distributions that are not necessarily easy to identify or express ("skewed, multi-modal, or otherwise complex distributions").
>
>   We really appreciate the useful pointers to related literature (see also below). However, with all due respect, we still believe that the nonparametric modeling of aleatoric uncertainty is generally very under-appreciated in model-based regression uncertainty related literature. Our main point of support for our belief is the list of survey papers we cite. To explicitly highlight this, we reformatted part of the related work section as an appendix (Appendix A.2.) where we survey 8 uncertainty-related survey papers for nonparametric approaches, including checking for MDN, Normalizing flows, MMD, and the energy score / CRPS / DISCONets.
>
>   However, we'd like to reiterate our appreciation for raising this issue. We did not realize that our text can indeed be interpreted in multiple places as stating that aleatoric uncertainty is under-researched *in general*, which we absolutely agree is not the case. We made attempts to clarify our intent in multiple places in the manuscript, please see the pdfdiff if possible. Further, we made many changes to more corretly represent prior nonparametric approaches:
>   * We removed the claim of our method being the first truly nonparametric way of learning aleatoric uncertainty
>   * We rephrased multiple sentences in the paper to more correctly represent prior non-Gaussian methods
>   * We extended the related work section to cite Ardizzone et al. 2019, Sendera et al., 2021, Depeweg et al. (2018)
>   * We highlight Bouchacourt et al.'s (2016) as the closest prior art to our work in the related work section.
> - We agree, see changes made above.
> - We agree. As also requested by Reviewer 59RD, we extend the appendix with a Section comparing against a simple unimodal gaussian regression model (MDN with 1 component, including deterministic and BNN versions). Regarding DISCO Nets and Depeweg et al. (2018), we are looking into the feasibility of experimental comparisons, and plan to further respond during the discussion period.
>
> __Questions__
>
> - Q1: We have extended the paper in multiple places (see above), further elaborating on the relation to prior work.
> - Q2: We agree that the networks we use to run the UCI benchmark are rather small (which is required for fair comparison against prior work). In general, we believe the approach to scale well to larger network sizes:
>   * The nondeterministic part only needs to be proportional to the complexity of the output distribution, rather than the input data, as the latter is first transformed using a traditional deterministic network into latent representations
>   * Even if the output distributions are assumed to be very complex, the synthetic Mona Lisa example is a good illustration that even a simple MLP network along with our method is able to represent a huge range of very complex distributions accurately. While the network we use for the Mona Lisa task is indeed larger than the ones used for the UCI benchmark, it is still relatively small compared to the amount of GPU computation modern deep learning approaches regularly use for processing complex input data.
>   We extended the paper with a condensed version of this argument at the end if the introduction section, replacing "limitations" with "scaling" and included a chart to demonstrate scaling in Appendix C.8. (please observe Figure 7 on page 26!)
>
>   Please let us know if we were able to answer your questions and whether you see any further issues that we should address.

---

> > ### Comment · Reviewer_cN1m · 2024-11-22
> >
> > Thanks for your answers, I look forward to the updated papers and experiments.
> > I think putting a greater emphasis on the multimodality compared to a generic one could really make it clear to the reader that this is where your method shines and is helpful.
> >
> > Regarding experiments. For me a comparison against Boucharcourt et al.'s DISCO net is sufficient during the rebuttal. Their approach should be rather straightforward to implement and run. Depeweg et al., might be somewhat more complex, and it is fine from my side if you are aware of it and explore the feasibility adding it in a camera-ready version. (I don't remember whether the authors ever provided a reference implementation for it that could quickly be adapted to current pytorch/jax/... )

---

> > > ### Author Response · Authors · 2024-11-27
> > >
> > > Thank you for your quick reply! We have extended the paper with appendices about DISCO Nets and Depeweg et al's approach.
> > >
> > > - For DISCO Nets, we measure performance on the original dataset of the paper (NYU Hand Pose) as well as the UCI benchmark in Appendix C.13. Our method seems to outperform the DISCO Nets method as expected (as with the correct parametrization and extensions, DISCO Nets essentially reproduces our simple un-weighted method in the univariate case).
> > > - For Depeweg et al, we fortunately found a very recent implementation which is co-authored by the Stefan Depeweg (cited as Lehman et al, 2024 in the newest revision of our paper), and managed to benchmark this implementation on the UCI datasets, please see Appendix C.14.
> > >
> > >   However we are still attempting to find a more stable set of hyperparameters, as our first attempts resulted in the method occasionally failing to converge on some train-test splits. If we manage to improve on the scores before the deadline of the camera-ready version, we will update the paper accordingly.
> > >
> > > We have updated the repo with our experiments regarding DISCO Nets. As for the experiments using BNN-LV, we are still working on a way to include the experiments in our code, which is somewhat challenging as the method always fails to converge in the latest versions of Lightning-UQ-Box. In fact the only way we were able to get the method to work was by taking the snapshot [at the time of the original pull request of BNN-LV](https://github.com/lightning-uq-box/lightning-uq-box/pull/21) and patching out a typo in the code where [X is used instead of y](https://github.com/lightning-uq-box/lightning-uq-box/blob/2a8f8c72e321ecb8e0a12b80e5993d2d284c6dbf/uq_method_box/uq_methods/bnn_vi_lv.py#L538C32-L538C48). However, it is nontrivial to refer to a working copy without breaking anonymity.

---

> > > > ### Comment · Reviewer_cN1m · 2024-11-27
> > > >
> > > > Thank you. With these latest experimental changes and reformulations in the main paper my concerns are resolved and I am recommending acceptance.
> > > >
> > > > (A very minor suggestion. I would remove the last two decimals from the SEM in Table 13. I know the current version is more faithful to Bouchacourt et al., but given the size of the first decimal the others are essentially random numbers without any significance)

---

> > > > > ### Author Response · Authors · 2024-12-03
> > > > >
> > > > > As today is the last day for comments, we just wanted to once again thank you for your valuable feedback and comments! We really appreciate the time and effort you put into reviewing our paper.
> > > > >
> > > > > (We agree with your points about SEM decimals and have made changes to the manuscript accordingly.)

---

### Official Review · Reviewer_yUKD · 2024-11-03

**Soundness:** 3
**Presentation:** 1
**Contribution:** 2
**Rating:** 5
**Confidence:** 3

**Summary:**

UQ is important for predictive modeling. Most existing methods assume parametric distributions around the true value, limiting their ability to capture skewed, multi-modal distributions.
The paper proposes a nonparametric, nondeterministic NN regression architecture that uses a loss function based on the continuous ranked probability score to capture the aleatoric uncertainty for systems with complex distributions.
The method is evaluated on a few benchmarks, showing better performance compared over other models.

**Strengths:**

The paper gives a decent introduction on the general UQ concepts and offers a clear summary of the relevant work.
To the reviewer's best judge, the authors seem to have a good grasp of the theory.

**Weaknesses:**

1. The nondeterministic NN presented in the paper is essentially a deterministic NN with a direct noise injection to the latent representation vector. The paper doesn’t provide a strong justification for why this approach would be effective, making the idea feel somewhat arbitrary.
2. To the best of the reviewer’s knowledge, some statements in the paper seem problematic. For example, "In other words, epistemic uncertainty can be interpreted as the uncertainty of aleatoric uncertainty." (Line 139, page 4).
3. It is unclear whether the method is practically useful. As presented in the paper, the method requires multiple inferences at each training and evaluation step, introducing additional computational cost. This cost is expected to increase for more complex target distributions, making the method less practical for complex systems with skewed, multi-modal non-Gaussian distributions—precisely the motivation behind this work.

**Questions:**

1. Could you clarify how injecting standard normal noise into the latent vector is expected to better capture aleatoric uncertainty?
2. Considering the higher computational costs reported compared to other methods, would this make it challenging to scale up to real, complex problems?

---

> ### Author Response · Authors · 2024-11-18
>
> Thank you for your review! We're pleased that you found the introduction and related work sections helpful. We respond to the listed weaknesses and questions in separate points below.
>
> __Weaknesses__
>
> - The nondeterminism of the networks is indeed achieved by evaluating the second part multiple times, with different latent noise samples as input. You are correct that there are multiple ways of injecting nondeterminism into a neural network, and that we do not present a comparison of these in the paper. However we do not believe this to be a crucial shortcoming of the paper, as the presented method is proven to outperform other approaches *as is*. Exploring more approaches is, however, an interesting area for further research. We made modifications to emphasize this fact in Section 4.3. (end of first paragraph).
> - Regarding *"In other words, epistemic uncertainty can be interpreted as the uncertainty of aleatoric uncertainty."*, can you please clarify what do you consider problematic?
>
>   We believe the statement to be true. For example in classic BNN-s, the parameters of the aleatoric uncertainty distribution are explicitly predicted by the nondeterministic network. In effect, the distribution of the parameters is determined by the nondeterminism of the BNN, therefore the quote exactly describes what happens in practice. Further, to support the argument in general, we point to two sources:
>   * In Bengs et al. (2023), the authors write (introduction, end of first paragraph):
>
>     > However, it does not allow the learner to express its epistemic uncertainty, namely, its lack of knowledge about how accurately $\tilde{p}$ approximates $p^*$. To capture this uncertainty as well, the learner is allowed to predict a second-order distribution $Q$. In other words, instead of committing to a single (point) prediction $\tilde{p}$, the learner assigns probabilities $Q(p)$ to all candidate distributions $p^*$.
>
>     (where in context $p^*$ is stated to capture aleatoric uncertainty).
>
>     Here, the authors explicitly state epistemic uncertainty to be a distribution over the candidate aleatoric distributions.
>   * In Amini et al. (2020), the authors propose modeling aleatoric uncertainty using the Normal distribution, and epistemic uncertainty as a parametric meta-distribution over the parameters of the aleatoric normal distribution (Normal-Inverse Gamma). This again explicitly highlights epistemic uncertainty as a meta-distribution over potential aleatoric distributions.
> - Indeed the sampling architecture results in increased running time compared to a deterministic network, however the method scales well to larger network sizes. To see this, do note that
>   * The nondeterministic part only needs to be proportional to the complexity of the output distribution, rather than the input data, as the latter is first transformed using a traditional deterministic network into latent representations
>   * Even if the output distributions are assumed to be very complex, the synthetic Mona Lisa example is a good illustration that even a simple MLP network along with our method is able to represent a huge range of very complex distributions accurately. While the network we use for the Mona Lisa task is indeed larger than the ones used for the UCI benchmark, it is still relatively small compared to the amount of GPU computation modern deep learning approaches regularly use for processing complex input data.
>   Further, as the UCI benchmark running times illustrate, the complexity of the proposed approach in practice is still much more favorable than the complexity of a BNN of comparable size.
>
>   Therefore, ultimately we do not believe running time to be a major concern regarding the applicability of the method.
>   We extended the paper with a condensed version of this argument at the end if the introduction section, replacing "limitations" with "scaling" and included a chart to demonstrate scaling in Appendix C.8. (please observe Figure 7 on page 26!)
>
> __Questions__
>
> - The nondeterminism of the network is necessary to be able to represent nonparametric output distributions through sampling, which is required for our method to model aleatoric uncertainty. Together with a sample-based loss (namely the CRPS), this makes the network capable of learning arbitrary aleatoric distributions, which we also demonstrate through the synthetic example in Section 5.1.
>
>   (Also, as we state in the first paragraph of Section 4.3, in theory any sufficiently capable nondeterministic network can in theory be used together with the CRPS loss, also discussed above in under *Weaknesses* nr. 1.)
> - Please see above under *Weaknesses* nr. 3.
>
> Please let us know if we were able to answer your questions and whether you see open issues that we should address.

---

> > ### Author Response · Authors · 2024-12-03
> >
> > Once again, thank you for your valuable feedback! We've carefully addressed your concerns in our response and the revised manuscript. As today is the last day for comments, we kindly ask you to consider raising your score based on these improvements. We truly appreciate your time and effort.

---

### Official Review · Reviewer_yrBT · 2024-11-04

**Soundness:** 3
**Presentation:** 3
**Contribution:** 3
**Rating:** 5
**Confidence:** 3

**Summary:**

The manuscript introduces 4 new architectures of nondeterministic neural networks for producing samples from the aleatoric target distribution. Furthermore, a new loss function for optimizing neural networks based on an empirical approximation of the "continuous ranked probability score" (CRPS) is introduced. Test over standard datasets for classification tasks as well as regressions tasks are discussed.

**Strengths:**

The idea is simple but rigorous and the implementation straightforward. The performance improvements, especially for the single-head weighted version with ensembling are small but consistent.

**Weaknesses:**

The presentation should be improved, see questions

**Questions:**

- The formulation of the problem at rows 92--96 should be more clear, in particular regarding the the definition of "small omega".
- at "The described setup provides, to our knowledge, the first nonparametric way to learn aleatoric uncertainty for neural network regression" rows 286-287. How do we know the uncertainty is aleatoric and not epistemic?
- Eq. 3: the loss is given by the difference between the cumulative distribution F of the labels for a batch and the stochastic results created by the network?
- the process of generating the different outcomes in figures 2 and 3 is not clear enough.
- In the benchmarking, are only 5 ensembles used? Is this shown / clear to be an ensemble large enough?
- could the process of generating the underlying distribution on the NLL (row 442) be briefly summarized? It is true this comes form literature, but a reminder could help the readability.

---

> ### Author Response · Authors · 2024-11-18
>
> Thank you for your review! We appreciate your kind words about our paper. We respond to the listed questions in separate points below.
>
> - Re lines 92-96: The variable $\omega$ here denotes an elementary event $\omega\in\Omega$, where $\Omega$ is the sample space of a probability field (usually denoted as $(\Omega, \mathcal{F}, P)$), see paragraph above.
> - Re lines 286-287: While our method can be argued to be capable of capturing epistemic uncertainty to some degree as well, the novelty we highlight is its nonparametric modeling of aleatoric uncertainty. The quoted sentence specifically refers to this capability.
>
>   Our method primarily captures aleatoric uncertainty because for any fixed input X, it learns to generate samples matching the ground truth distribution $P(Y\mid X)$, independent of model uncertainty. Unlike epistemic uncertainty which reflects model ignorance, our sampling approach directly targets the inherent data variability at each X.
> - Re eq. 3: As part of the background section, Equation 3 describes the CRPS score in general, which is an analytic score function to compare the similarity of two arbitrary univariate distributions, and assumes exact knowledge of the CDF of predicted distribution $F$ and a sample from the ground-truth distribution $y$. As such, considerations around batches or networks are not directly applicable at this point.
>
>   Later (eq. 5), we introduce the sample-based approximation of the CRPS. Here the formula indeed compares the stochastic results produced by the network to the ground truth $y$ value. In particular, $F$ from eq.3. is approximated by the empirical CDF of the samples drawn as stochastic outputs from the network.
> - Re figures 2 and 3: We appreciate the comment! We made an attempt to describe the process more clearly by extending the Figure captions.
> - Re 5 ensembles: We chose the number as 5 following the choices made by the original paper (Lakshminarayanan et al. (2017)). Note that increasing the number of ensembled networks greatly increases the size of the model, as well as substantially increases the runtime of Lakshminarayanan et al. (2017).
> - Re line 442: The methods we compare against express the aleatoric likelihood as parametric distributions, and therefore are able to compute the likelihood analytically. We made changes to highlight this fact in Section 5.3.1. (last two lines, cca. 451-452).
>
> Please let us know if we were able to answer your questions and whether you see open issues that we should address.

---

> > ### Author Response · Authors · 2024-12-03
> >
> > Once again, thank you for your valuable feedback! We've carefully addressed your concerns in our response and the revised manuscript. As today is the last day for comments, we kindly ask you to consider raising your score based on these improvements. We truly appreciate your time and effort.

---

### Official Review · Reviewer_59RD · 2024-11-08

**Soundness:** 3
**Presentation:** 3
**Contribution:** 3
**Rating:** 8
**Confidence:** 3

**Summary:**

This paper presents an for modeling aleatoric uncertainty in neural network regression tasks using a nondeterministic architecture optimized with loss functions derived from the continuous ranked probability score (CRPS). The method enables learning arbitrary output distributions in a nonparametric way by training the network to generate samples from the target's aleatoric distribution and the training the underlying model for regression. The authors present interesting results on inverse MNIST and a set of synthetic and non-synthetic UCI datasets.

**Strengths:**

- I like the fact that this paper focuses on aleatoric uncertainty as opposed to total uncertainty or epistemic uncertainty. Aleatoric uncertainty has been largely overlooked

- The architecture choice is very interesting and intuitive and the results presented are very useful. The loss function is also described well theoretically and it's nice to see it build upon existing literature for a new task.

- The paper presents extensive experiments on both synthetic and real-world datasets, including univariate and multivariate problems.

**Weaknesses:**

- Heteroscedatic regression where we explicitly model the prediction and the uncertainty around it is very related to this line of work. However, the authors do not describe why should one prefer that line of work versus their approach. See [1] for an example on heteroscedatic regression.

- While the authors do acknowledge it, the current method can be computationally expensive. This might be problematic as we try to scale larger architectures.

References

[1] Korattikara, A., Rathod, V., Murphy, K. and Welling, M., 2015. Bayesian dark knowledge. arXiv preprint arXiv:1506.04416.

**Questions:**

- Isn't heteroscedatic regression also modeling aleatoric uncertainty by explicitly putting an uncertainty on the outputs?

- I think it'll be useful if you also explain why are we focused on cumulative distribution and expanding section 2.4 so that it's more accessible to readers.

- Can you talk about how we would scale this to larger models?

- Can you also talk how about how this method would saturate with # of samples drawn?

---

> ### Author Response · Authors · 2024-11-18
>
> Thank you for your review! We were delighted to read that you found our manuscript and its contents interesting, intuitive, and useful. We respond to the listed weaknesses and questions in separate points below.
>
> __Weaknesses__
>
> - Indeed heteroscedastic regression is a very relevant concept here. In fact, any model which predicts a non-constant variance can be considered a heteroscedastic regression model, which includes all the models considered in the paper, including both newly introduced methods and the baselines. However, if we restrict heteroscedastic regression to mean explicitly predicting two parameters (mean and variance), then the paper can indeed be considered lacking in comparisons. Do note, however, that one of the baselines *is* in fact such a model (PBP-MV, which is an advanced type of Bayesian neural network, and outputs the aleatoric distribution in the form of two parameters for an assumed Gaussian distribution).
>
>   We agree that more direct comparisons against simple examples of heteroscedastic regression can serve as a useful additional context in the paper. Accordingly, we are extending the manuscript with a new appendix with two additional measurements: a simple heteroscedastic neural network regression model and a classic BNN using Variational Inference, see Appendix C.11.
>
> - Indeed the sampling architecture results in increased running time compared to a deterministic network, however the method scales well to larger network sizes. To see this, do note that
>     * The nondeterministic part only needs to be proportional to the complexity of the output distribution, rather than the input data, as the latter is first transformed using a traditional deterministic network into latent representations
>     * Even if the output distributions are assumed to be very complex, the synthetic Mona Lisa example is a good illustration that a simple MLP network along with our method is able to represent a huge range of very complex distributions accurately. While the network we use for the Mona Lisa task is indeed larger than the ones used for the UCI benchmark, it is still relatively small compared to the amount of GPU computation modern deep learning approaches regularly use for processing complex input data.
>
>   Further, as the UCI benchmark running times illustrate, the complexity of the proposed approach in practice is still much more favorable than the complexity of a BNN of comparable size.
>
>   We extended the paper with a condensed version of this argument at the end if the introduction section, replacing "limitations" with "scaling" and included a chart to demonstrate scaling in Appendix C.8. (please observe Figure 7 on page 26!)
>
> __Questions__
>
> - Re heteroscedastic regression: See above when discussing weaknesses
> - Re CDF: We agree that more clarity could have been used here. We have expanded this section as much as the page constraints allow.
> - Re scaling: See above when discussing running time
> - Re saturation with samples: Very thoughtful question! To respond, we introduce a new appendix to compare runs with different numbers of samples drawn during training, presenting results in running time and prediction accuracy. See Appendix C.12.
>
> Please let us know if we were able to answer your questions and whether you see open issues that we should address.

---

> > ### Comment · Reviewer_59RD · 2024-11-27
> >
> > I thank the authors for their response and doing the additional experimentation. I'd encourage the authors to surface the insights from the additional experiments into their main body for the next version of the manuscript.
> >
> > I'll be raising my score since I am happy with the changes the authors have made.

---

> > > ### Author Response · Authors · 2024-12-03
> > >
> > > As today is the last day for comments, we just wanted to once again thank you for your valuable feedback and comments! We really appreciate the time and effort you put into reviewing our paper.
> > >
> > > (As for the suggestion, we agree, and will further attempt to include conclusions from Appendix C.11 into the main text for the next revision, notably highlighting the simple BNN as the dominant method for the *concrete* dataset.)

---

### Meta-Review · Area_Chair_NsYG · 2024-12-22

**Metareview:**

The paper proposes a nondeterministic neural network regression architecture optimized using a loss function based on the continuous ranked probability score to capture aleatoric uncertainty in a nonparametric manner. The reviewers appreciated the novelty of focusing on the nonparametric modelling of aleatoric uncertainty, the idea and architecture choice being intuitive and rigorous, the scalability of the approach, and the extensive experimental validations, including new comparisons introduced during the rebuttal period. The reviewers disagree on the clarity of the paper, with some finding the paper well-written and easy to follow and others saying that the presentation should be improved. Concerns were raised regarding the incomplete framing of related literature, potential scalability issues to large models, and the lack of baseline comparisons in the initial submission (DISCO net). Most issues were addressed in the rebuttal and discussions through refined framing of the paper’s contributions, additional experiments and clarifications on, e.g., scalability. After the discussion and revisions, two reviewers recommended accepting the paper, while two reviewers still found the papers marginally below acceptance.  Based on the significant strengths of the paper, the revisions, and the consensus on its contributions, I recommend accepting the paper.

**Additional Comments On Reviewer Discussion:**

The discussion and changes are described in the meta-review.

---

### Decision · Program_Chairs · 2025-01-22

Accept (Poster)